# SPARSE-COMPRESSION DIFFUSION MODELS

## ABSTRACT

Diffusion models have demonstrated capabilities in generating synthetic data, especially the pictural ones. However, the conventional reverse diffusion process merely overfits in data denoising. Empirical studies indicate that the denoising mechanism obstructs the diffusion models to generate robust physical synthetic data. In this work, we propose the Sparse-Compression Diffusion Models (SCDM), with a novel reverse diffusion process that enforces compression. The compression attempts to recover the low-dimensional and sparse representations, potentially enabling the ability to capture the latent factors of some rules. SCDM demonstrates great generalization on datasets of images and physical scenarios. Our experiments also prove that the sparse latent representations learned by SCDM align well with scientific theories in physical scenarios.

## 1 INTRODUCTION

Diffusion models have emerged as a powerful class of generative models to generate synthetic samples from the learned data distributions, advancing the data synthesis in the field of computer vision Ho et al. (2020); Song et al. (2020). These successes suggest the great potential to build a data recovery model along with the denoising process. However, a conventional reverse diffusion process merely overfits a data denoising objective. In the scenarios where the data to be recovered are from real-world observations, the conventional denoising objective function would omit the possibility to align with the latent factors of some rules. As shown in Figure 1, we demonstrate the generation results by a representative diffusion model (DDPM) Ho et al. (2020) and our Sparse-Compression Diffusion Models (SCDM).

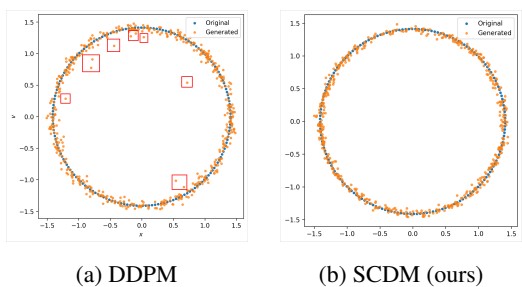

|             |              |
|:-----------:|:------------:|
| (a) DDPM    | (b) SCDM (ours) |

Figure 1: Generative models (DDPM vs. our SCDM) trained on a structured dataset. The red boxes mark the improperly generated synthetic samples. DDPM exhibits degraded quality due to its naive denoising objective, whereas our proposed method (SCDM) generates better samples.

The underlying physical processes in nature are typically governed by a small number of latent factors and interactions Schmidt & Lipson (2009); Brunton et al. (2016). For example, a swinging pendulum may produce rich sensor data, but its dynamics reduce to two variables governed by a second-order ordinary differential equation. For scientists to find these concise yet powerful analytical latent factors, theories are derived from observations, which subsequently guide the induction of hypothesis and further inferences. In this work, we explore the possibility to train a generative model that is implicitly restricted by the latent factors, in imitation of the scientists' induction and inference process. The learned generative model could be further used to generate robust and reliable synthetic datasets.

We propose the Sparse-Compression Diffusion Models (SCDM), which integrates diffusion models with sparse representation learning via sparse rate reduction. In contrast to the conventional diffusion models, the SCDM introduces a *compression* objective during the reverse diffusion process. The intuition is that the diffused data should be denoised under some compact and intrinsic assumptions, so that the recovered data could be robust. Our SCDM leverages the conventional diffusion model's

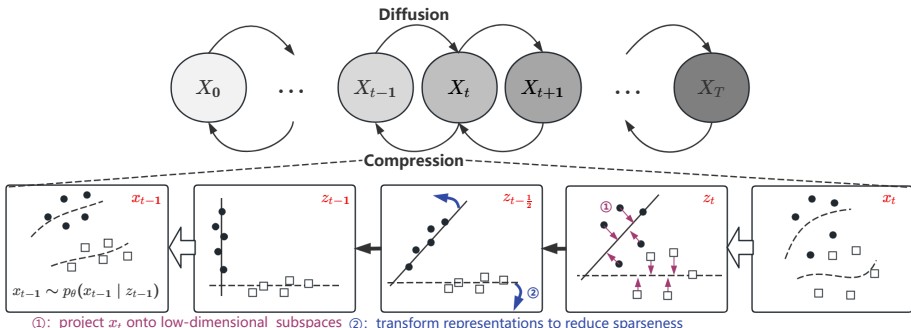

Figure 2: Overview of the process in the Sparse-Compression Diffusion Models (SCDM). The forward arrow illustrates the diffusion process, where $x_0$ is gradually corrupted into $x_T$ by successive noise injection. The backward arrow illustrates the compression (denoising) process, where $x_t$ is refined step-by-step via a guided projection function. The compression process recovers sparse representations: $x_t \rightarrow z_t \rightarrow z_{t-\frac{1}{2}} \rightarrow z_{t-1}$, guiding the denoising step $x_{t-1} \sim p_\theta(x_{t-1} \mid z_{t-1})$.

capabilities of data recovery, while simultaneously aligning with scientific hypotheses of the latent factors.

Figure 2 illustrates the SCDM, which consists of two processes: a conventional forward diffusion process that progressively adds noise to original data $x_0 \rightarrow x_T$, and a novel compression process that iteratively removes noise by enforcing structured sparsity. Specifically, during the compression process, we first encode $x_t$ to obtain a unified embedding $z_t$. Then we project $z_t$ onto some low-dimensional subspaces to capture relevant components, generating $z_{t-\frac{1}{2}}$; and applying a soft threshold , producing sparse $z_{t-1}$. Finally, $z_{t-1}$ is used to guide the denoising step $x_{t-1} \sim p_\theta(x_{t-1} \mid z_{t-1})$ in the reverse diffusion process.

The contributions of this paper are two-fold:

- We introduce the Sparse-Compression Diffusion Models (SCDM), a novel framework that integrates diffusion modeling with sparse rate reduction, automatically denoising data and simultaneously extracting compressed and low-dimensional factors from physical data.

- Empirical results demonstrate that the sparse latent representations learned by SCDM align well with latent factors identified by theories, which substantially leads to the robustness of diffusion models in physical data synthesis. SCDM also performed well on CIFAR-10, especially when the training data was compromised.

## 2 RELATED WORK

### 2.1 DIFFUSION MODEL

Recent work has explored architectural improvements to enhance performance and efficiency. Peebles & Xie (2023) proposed Diffusion Transformers (DiT), replacing the U-Net backbone with pure transformers to scale model capacity and fidelity. In contrast, Tian et al. (2025) introduced Diffusion CNN (DiC), a convolutional alternative that achieves faster sampling and strong visual quality using optimized 3×3 convolutions and sparse skip connections. Incorporating domain-specific priors into diffusion models has also gained attention. Shu et al. (2023) developed physics-informed diffusion models for fluid dynamics by embedding PDE constraints and coarse resolution inputs into the U-Net decoder, improving robustness and physical consistency. Further, Williams et al. (2023) provided a theoretical framework analyzing U-Net encoder-decoder dynamics, showing that high-frequency signals rapidly decay under diffusion noise. Their findings motivate architectural variants like Multi-ResNets, which incorporate structured priors and basis functions to enhance decoder fidelity.

## 2.2 GENERATIVE MODELS WITH EXPLICIT SPARSE REGULARIZATION

Sparsity constraints have been effectively integrated into generative models like VAEs, GANs, and diffusion models to enhance interpretability and performance. Moran et al. (2021) propose Sparse VAE, using sparse decoding to achieve identifiable latent representations. Their approach enforces structured sparsity through sparse weight matrices, leading to improved disentanglement and lower reconstruction errors. Xiao et al. (2025) introduce SC-VAE, which combines traditional sparse coding with variational autoencoders via a learned ISTA encoder. SC-VAE produces sparse and interpretable basis representations while achieving superior reconstruction quality compared to standard VAEs. Sparse Data Diffusion (SDD) by Ostheimer et al. (2025) addresses inherently sparse datasets by introducing discrete Sparsity Bits to explicitly encode zero patterns, effectively preserving structural sparsity in scientific and biological data. Ganz & Elad (2022) integrate sparsity-inducing regularizer into GAN architectures, interpreting GAN generators as multi-layer sparse coding systems. Their experiments show that sparse activation layers significantly improve image synthesis quality and performance in denoising tasks.

## 2.3 SPARSE CODING

Sparse coding, first introduced by Olshausen & Field (1996), models data as sparse linear combinations over overcomplete dictionaries. Efficient learning algorithms such as K-SVD Aharon et al. (2006) and online dictionary learning Mairal et al. (2009) made sparse coding scalable and practical. Classical methods like SRC Wright et al. (2009) and SSC Elhamifar & Vidal (2013) demonstrated strong performance in classification and clustering tasks. Sparse coding principles were later integrated into deep learning, with optimization algorithms like ISTA reinterpreted as trainable neural layers Gregor & LeCun (2010). Probabilistic models Kingma & Welling (2014); Higgins et al. (2017) further emphasized reducing redundancy to promote disentangled and interpretable representations. Recent work combines sparse coding with rate-distortion theory in deep architectures Chan et al. (2022); Yu et al. (2023), yielding white-box models with improved compression and interpretability.

## 2.4 SPARSE LATENT FACTORS IN SCIENTIFIC DOMAINS

Empirical studies consistently highlight that the many fields in nature can be described by concise latent factors. Human motion capture data exhibit intrinsic sparsity, efficiently captured by PCA or deep generative models with few latent dimensions. Physical dynamical systems, as demonstrated by Brunton et al. (2016), are effectively described by a limited set of active variables. Neuroscience and biology data Safonova et al. (2004) similarly reveal latent factors: neural recordings and gene expression profiles can be explained by few active factors like neural ensembles or gene modules. Robotics and sensor data Rhodes (2024); Carvalho et al. (2007) frequently lie on low-dimensional manifolds reflecting underlying physical degrees of freedom, supporting sparse representation approaches.

These findings underline why sparse priors have become essential for generative and representation learning. However, current diffusion models usually ignore the potential structure of the data during the reverse diffusion process. To address this limitation, we suggest integrating explicit sparsity into the reverse process of DDPM.

## 3 PRELIMINARY

### 3.1 DENOISING DIFFUSION PROBABILISTIC MODELS

Denoising Diffusion Probabilistic Models (DDPM) Ho et al. (2020) are a class of generative models, which consists of a forward and a reverse diffusion process. In the forward process, Gaussian noise is gradually added to the clean data $x_0$ over $T$ timesteps, forming a Markov chain:

$$q(\mathbf{x}_{1:T} \mid \mathbf{x}_0) = \prod_{t=1}^{T} q(\mathbf{x}_t \mid \mathbf{x}_{t-1}), \tag{1}$$

where $q(\mathbf{x}_t \mid \mathbf{x}_{t-1}) = \mathcal{N}(\mathbf{x}_t; \sqrt{\alpha_t}\,\mathbf{x}_{t-1}, \beta_t\mathbf{I})$, $\alpha_t$ and $\beta_t$ denote the variance-preserving noise schedule parameters. The reverse process aims to denoise $x_t$ back to $x_0$ by learning a noise prediction network $\epsilon_\theta(x_t, t)$. The training objective minimizes the discrepancy between the true noise $\boldsymbol{\epsilon} \sim \mathcal{N}(\mathbf{0}, \mathbf{I})$ and the predicted noise, formulated as:

$$\mathcal{L} = \mathbb{E}_{\mathbf{x}_0,\, \boldsymbol{\epsilon},\, t} \left[ \left\| \boldsymbol{\epsilon} - \boldsymbol{\epsilon}_\theta \left( \sqrt{\bar{\alpha}_t}\,\mathbf{x}_0 + \sqrt{1 - \bar{\alpha}_t}\,\boldsymbol{\epsilon},\, t \right) \right\|^2 \right], \tag{2}$$

where $\bar{\alpha}_t = \prod_{s=1}^{t} \alpha_s$ is the cumulative product of noise schedules. After training, data generation is performed by iteratively sampling from the learned reverse transitions.

The latent variable $x_t$ serves two roles in the diffusion process. On the one hand, it originates from a forward encoding of $x_0$ and thus carries task-relevant information entangled with noise. On the other hand, it acts as the initial condition for reconstructing $x_0$ during reverse denoising. We hope to find a representation that can extract the explanatory controlling factor describing $x_0$ from the reverse process.

## 3.2 SPARSE RATE REDUCTION

Recent advances in representation learning suggest that effective representations should be both compressed and sparse, capturing the essential structure of data while discarding redundancy. The sparse rate reduction framework (CRATE) Yu et al. (2023) formalizes this principle by proposing a unified objective that simultaneously minimizes the lossy coding rate of token representations and promotes sparsity. Under the assumption that each token representation $z_i \in Z$ lies in a mixture of low-dimensional Gaussian subspaces supported on orthonormal bases $\{U_j\}_{j=1}^{N}$, the representation function $f$ maps input $X$ to $Z = f(X)$, optimized via:

$$\max_{f \in \mathcal{F}} \mathbb{E}_Z \left[ R(Z) - R^c(Z; U^{[N]}) - \lambda \|Z\|_0 \right], \tag{3}$$

where $R(Z)$ denotes the total coding cost of $Z$, $R^c(Z; U^{[N]})$ is the rate under a learned mixture of $N$ low-dimensional Gaussian subspaces, and $\|Z\|_0$ encourages sparsity. The detailed formulations of $R(Z)$ and $R^c(Z; U^{[N]})$ are provided in equations 8 and 10. This objective encourages representations that are informative yet parsimonious, aligning with principles from classical sparse coding and information theory.

To effectively realize the above objective, the representation function is designed as a composition of multiple sparse transformation layers. Each layer is designed to perform two key operations: (1) The compression objective, corresponding to minimizing the projected coding rate $R^c$, is implemented via Multi-Subspace Self-Attention (MSSA), which softly projects representations onto multiple orthogonal subspaces using attention weights derived from local coding cost; and (2) The sparsity objective, encouraging low $|z_t|_0$, is enforced through the Iterative Shrinkage-Thresholding Algorithm (ISTA), which performs soft-thresholding on the projected representations to retain only salient components. In the next section, we extend this principle into the diffusion framework. The compression–sparsity objective is imposed at each reverse diffusion step on the latent $z_t$, jointly optimized with the denoising objective. This dynamic integration, together with subspace projection and ISTA iterations, distinguishes our SCDM from CRATE and constitutes our main methodological novelty.

# 4 METHODOLOGY

## 4.1 MODEL FRAMEWORK

From the forward process

$$x_t = \sqrt{\bar{\alpha}_t}\,x_0 + \sqrt{1 - \bar{\alpha}_t}\,\varepsilon, \qquad \varepsilon \sim \mathcal{N}(0, I), \tag{4}$$

we have

$$\varepsilon = \frac{x_t - \sqrt{\bar{\alpha}_t}\,x_0}{\sqrt{1 - \bar{\alpha}_t}}, \tag{5}$$

and therefore, taking the conditional expectation given $x_t$ yields

$$
\mathbb{E}\big[\varepsilon \mid x_t\big] = \mathbb{E}\Big[\frac{x_t - \sqrt{\bar{\alpha}_t}\, x_0}{\sqrt{1 - \bar{\alpha}_t}} \,\Big|\, x_t\Big]
$$

$$
= \frac{1}{\sqrt{1 - \bar{\alpha}_t}}\Big(x_t - \sqrt{\bar{\alpha}_t}\, \mathbb{E}[x_0 \mid x_t]\Big). \tag{6}
$$

The quantity of interest is precisely the structural information in $x_t$ that determines the posterior mean $\mathbb{E}[x_0 \mid x_t]$. In other words, the standard objective drives $\hat{\varepsilon}_\theta(x_t, t)$ towards the Bayes-optimal predictor $\mathbb{E}[\varepsilon \mid x_t]$ in the mean squared sense. Consequently, as long as a representation extracted from $x_t$ preserves the $x_0$-like structural components that govern $\mathbb{E}[x_0 \mid x_t]$, it is in principle sufficient for accurate $\varepsilon$-prediction as well.

In this work, we follow the sparse rate reduction principle, which encourages each variable $x_t \in \mathbb{R}^{D \times n}$ in the compression process to be both compact and sparse. For clarity, we define $K$ as the number of ISTA iterations performed at each timestep, and $U = \{U^{(j)}\}_{j=1}^N$ as a bank of $N$ learned subspace bases used for feature projection. Each basis $U^{(j)} \in \mathbb{R}^{D \times p}$ is shared across timesteps and facilitates compression and sparsification. We present SCDM's procedure in Algorithm 1.

Specifically, at each timestep t, given a noisy latent variable $x_t$, we first encode it into a unified representation space $z_t$ via an NN $T$:

$$
z_t = T(x_t). \tag{7}
$$

We measure the compactness of these learned representations using the coding rate:

$$
R(z_t) \triangleq \frac{1}{2} \log \det \Big(\mathbf{I} + \frac{1}{\epsilon^2}\, z_t z_t^\top\Big). \tag{8}
$$

where the parameter $\epsilon$ ensures that the mean squared error tolerance. The above coding rate reflects the average number of bits required to compress $z_t$ within an error tolerance $\epsilon$.

Projecting $z_t$ onto the subspaces defined by $U$, we obtain the total encoding rate:

$$
R^c\big(z_t;\, U^{[N]}\big) = \sum_{j=1}^{N} R\big(U^{(j)\top} z_t\big) \tag{9}
$$

$$
= \sum_{j=1}^{N} \log \det \Big(\mathbf{I} + \frac{p}{2n\varepsilon^2}\, \big(U^{(j)\top} z_t\big)^\top \big(U^{(j)\top} z_t\big)\Big). \tag{10}
$$

This projected coding rate quantifies the complexity of representation features within each subspace. A lower projected coding rate indicates that the latent representation is captured more efficiently within the subspaces, reflecting better compression and less redundancy.

To encourage efficient **compression**, we aim to reduce the total coding cost of the representation by projecting it onto a structured subspace. Specifically, we formulate the objective as maximizing the compression gain:

$$
\max_{z_t} \Delta R(z_t; U^{[N]}) = R(z_t) - R^c(z_t; U^{[N]}). \tag{11}
$$

To further encourage **sparsity**, we impose a penalty that enforces the representation to remain sparse in the standard coordinate basis. This yields the final training objective for learning representations that are both subspace-aware and sparsity-enhanced:

$$
\max_{f \in \mathcal{F}} \mathbb{E}_{z_t} \Big[\Delta R(z_t; U^{[N]}) - \lambda \|z_t\|_0\Big]
$$

$$
= \max_{f \in \mathcal{F}} \mathbb{E}_{z_t} \Big[R(z_t) - R^c(z_t; U^{[N]}) - \lambda \|z_t\|_0\Big], \tag{12}
$$

where $z_t = f(x_t)$ denotes the encoded representation at timestep $t$ and $\lambda$ controls the sparsity penalty. The above sparse program is made trainable by replacing $\|z_t\|_0$ with its convex surrogate $\|z_t\|_1$ and adopting the unconstrained LASSO form:

$$
\max_{f \in \mathcal{F}} \mathbb{E}_{z_t}\big[\Delta R(z_t; U^{[N]}) - \lambda \|z_t\|_1\big]. \tag{13}
$$

---

**Algorithm 1** Training and Sampling Procedure of SCDM

---

1: **Training Phase**
2: Sample $x_0 \sim q(x_0)$; $t \sim \text{Uniform}\{1, \ldots, T\}$; $\varepsilon \sim \mathcal{N}(0, \mathbf{I})$     $\triangleright$ Draw data, timestep, and Gaussian noise
3: $x_t \leftarrow \sqrt{\bar{\alpha}_t}\, x_0 + \sqrt{1 - \bar{\alpha}_t}\, \varepsilon$     $\triangleright$ Forward diffusion to obtain noisy sample
4: $z_t^0 \leftarrow T(x_t)$     $\triangleright$ Transform into representation space
5: **for** $k = 1$ **to** $K$ **do**
6:     $z_{t-\frac{1}{2}}^k \leftarrow \text{Proj}\big(z_t^{k-1}\big)$     $\triangleright$ Projection onto low-dimensional subspace using 14
7:     $z_{t-1}^k \leftarrow \text{Soft}\Big(z_{t-1}^{k-1} - \eta_k \nabla_z \big\| z_{t-\frac{1}{2}}^k - z_{t-1}^{k-1} \big\|_2^2, \lambda\Big)$     $\triangleright$ ISTA-style update using 16
8: **end for**
9: $\hat{\varepsilon}_\theta \leftarrow g_\theta\big(T^{-1}(z_{t-1}^K), t\big)$     $\triangleright$ Noise prediction network
10: $\theta \leftarrow \theta - \eta \nabla_\theta \|\varepsilon - \hat{\varepsilon}_\theta\|_2^2$     $\triangleright$ Parameter update by gradient descent
11: **Sampling Phase**
12: $x_T \sim \mathcal{N}(0, \mathbf{I})$     $\triangleright$ Initialize from Gaussian prior
13: **for** $t = T$ **to** $1$ **do**
14:     $z_t^0 \leftarrow T(x_t)$     $\triangleright$ Transform into representation space
15:     **for** $k = 1$ **to** $K$ **do**
16:        $z_{t-\frac{1}{2}}^k \leftarrow \text{Proj}\big(z_t^{k-1}\big)$     $\triangleright$ Projection onto low-dimensional subspace using 14
17:        $z_{t-1}^k \leftarrow \text{Soft}\Big(z_{t-1}^{k-1} - \eta_k \nabla_z \big\| z_{t-\frac{1}{2}}^k - z_{t-1}^{k-1} \big\|_2^2, \lambda\Big)$     $\triangleright$ ISTA-style update using 16
18:     **end for**
19:     $\hat{\varepsilon}_\theta \leftarrow g_\theta\big(T^{-1}(z_{t-1}^K), t\big)$     $\triangleright$ Predict current noise using 17
20:     $\gamma \sim \mathcal{N}(0, \mathbf{I})$ **if** $t > 1$ **else** $\gamma \leftarrow 0$     $\triangleright$ Stochastic noise for reverse step
21:     $\mu_\theta \leftarrow \dfrac{1}{\sqrt{\bar{\alpha}_t}}\Big(T^{-1}(z_{t-1}^K) - \dfrac{\beta_t}{\sqrt{1 - \bar{\alpha}_t}}\, \hat{\varepsilon}_\theta\Big)$     $\triangleright$ Mean estimate of reverse process using 19
22:     $x_{t-1} \leftarrow \mu_\theta + \sigma_t \gamma$     $\triangleright$ One-step denoising update
23: **end for**
24: **return** $x_0$

---

To effectively realize the above objective, the representation function is implemented as a composition of $K$ sparse transformation layers, following the sparse rate reduction principle. At each layer $k$, the latent vector is first softly projected onto a mixture of orthogonal subspaces, where the projections are weighted according to their local activation strengths:

$$z_{t-\frac{1}{2}}^k = \text{Proj}\big(z_t^{k-1}\big) = \sum_{j=1}^N \omega_k^{(j)}\big(z_t^{k-1}\big)\, U^{(j)} U^{(j)^\top} z_t^{k-1}, \tag{14}$$

$$\omega_k^{(j)}\big(z_t^{k-1}\big) = \frac{\exp\big(\|U^{(j)^\top} z_t^{k-1}\|_2^2\big)}{\displaystyle\sum_{\ell=1}^N \exp\big(\|U^{(\ell)^\top} z_t^{k-1}\|_2^2\big)}. \tag{15}$$

where the weights $\omega_k^{(j)}$ favor subspaces whose projections retain higher energy (i.e., larger projected norm), effectively emphasizing the most informative directions for compression.

The compressed representation is then refined by an ISTA-style soft-threshold update,

$$z_{t-1}^k = \text{Soft}\Big(z_{t-1}^{k-1} - \eta_k \nabla_{z_{t-1}} \tfrac{1}{2} \big\| z_{t-\frac{1}{2}}^k - z_{t-1} \big\|_2^2 \Big|_{z_{t-1} = z_{t-1}^{k-1}}, \eta_k \lambda\Big), \tag{16}$$

where $\text{Soft}(x, \tau) = \text{sign}(x) \max(|x| - \tau, 0)$ and we initialize $z_{t-1}^0 = z_t$. Here $\eta_k$ is the step size for proximal gradient descent on the smooth term $\frac{1}{2}\|z_{t-\frac{1}{2}}^k - z_{t-1}\|_2^2$, and the shrinkage threshold is $\tau = \eta_k \lambda$. Larger $\lambda$ induces stronger sparsity.

Each layer performs a softmax-weighted projection onto learned orthogonal subspaces, followed by an ISTA-style sparsification, jointly optimizing the compression–sparsity objective in Eq. equation 12. Stacking $K$ such layers yields compact, informative, and interpretable sparse representation.

The sparse representation $z_{t-1}^K$ is subsequently used to predict the component $\hat{\varepsilon}\theta(x_t, t)$ through a decoder network $g_\theta$:

$$\hat{\varepsilon}_\theta(x_t, t) = g_\theta(T^{-1}(z_{t-1}^K), t). \tag{17}$$

where $T^{-1}$ is implemented through an neural network to map $z_{t-1}^K$ to same ambient space as $x_t$. This noise estimate is then used to generate the next sample $x_{t-1}$ in the chain according to the standard DDPM reverse sampling rule:

$$x_{t-1} \sim p_\theta(x_{t-1} \mid z_{t-1}) = \mathcal{N}\left(\mu_\theta(z_{t-1}^K, t), \sigma_t^2 \mathbf{I}\right), \tag{18}$$

$$\mu_\theta(z_{t-1}^K, t) = \frac{1}{\sqrt{\alpha_t}} \left( T^{-1}(z_{t-1}^K) - \frac{\beta_t}{\sqrt{1 - \bar{\alpha}_t}} g_\theta(T^{-1}(z_{t-1}^K), t) \right). \tag{19}$$

## 5 EXPERIMENT

### 5.1 IMAGE GENERATION WITH SCDM

To evaluate the robustness of SCDM under noisy training data, we conduct a experiment on CIFAR-10. We construct multiple corrupted training sets by randomly selecting a certain proportion of training images (Noise = 0%, 30%, 40%, 50%, 60%) and adding zero-mean Gaussian noise with standard deviation $\sigma$ in pixel space, followed by clipping to [0,1]. The Inception Score and FID are computed with clean image set. A visualization of the clean vs 50% corrupted training data is deferred to the Appendix (Fig. 6).

**Baselines.** **DDPM** Ho et al. (2020) is the standard denoising diffusion model that serves as our main reference. **SCVAE** Xiao et al. (2025) is a variational auto-encoder with LISTA-style sparse encoders, providing a non-diffusion sparsity baseline.

We follow the standard DDPM framework for CIFAR-10 (same U-Net backbone, time-embedding strategy, and linear $\beta_t$ schedule from $\beta_1 = 10^{-4}$ to $\beta_T = 0.02$ with $T = 1000$ steps, predicting $\varepsilon_\theta(x_t, t)$ with an $\ell_2$ loss). We implement the projection operator in Algorithm 1 using a bank of $1 \times 1$ analysis–synthesis convolutions that define $N = 8$ low-rank channel subspaces (each of rank $r = 32$). We stack K=3 such projection–shrinkage layers to yield the sparse representation $z_t$.

The results are summarized in Table 1. We additionally present visual examples of generated samples under different noise levels in Appendix E. SCDM delivers strong performance across all settings, and its advantage becomes even more pronounced when noise is injected into the training data. This behavior suggests that the combination of compression and sparsity effectively steers the generative process toward cleaner underlying structures, thereby substantially improving robustness under noisy training conditions. During training, we monitor the training loss given by the EMA-smoothed $x_0$ reconstruction MSE, as shown in Appendix (Fig. 11). Note that the model is still optimized with the standard $\varepsilon$-prediction objective on $\hat{\varepsilon}_\theta(x_t, t)$ as defined in equation 17, rather than an $x_0$ reconstruction loss.

To understand the role of each component in SCDM, we conduct ablations on the 30% noisy CIFAR-10 dataset by isolating (i) the low-dimensional projection in 14 and (ii) the ISTA-style sparsification in 16. The projection-only variant is denoted SCDM-P, and the ISTA-only variant is denoted SCDM-I.

Table 1: FID and IS on CIFAR-10 under different levels of noisy training data.

| Noise (%) | Model | FID ↓ | IS ↑ |
|---|---|---|---|
| 0 | SCDM | **2.90** | **9.92 ± 0.16** |
| 0 | DDPM | 3.17 | 9.46 ± 0.11 |
| 0 | SC-VAE | 4.41 | 6.22 ± 0.08 |
| 30 | SCDM | **23.96** | **8.97 ± 0.14** |
| 30 | DDPM | 30.55 | 8.77 ± 0.08 |
| 30 | SC-VAE | 45.93 | 5.99 ± 0.07 |
| 40 | SCDM | **35.56** | **8.57 ± 0.16** |
| 40 | DDPM | 43.90 | 8.15 ± 0.13 |
| 40 | SC-VAE | 49.01 | 5.79 ± 0.1 |
| 50 | SCDM | **53.07** | **7.72 ± 0.10** |
| 50 | DDPM | 59.67 | 7.44 ± 0.11 |
| 50 | SC-VAE | 53.11 | 5.764 ± 0.08 |
| 60 | SCDM | **61.09** | **7.37 ± 0.06** |
| 60 | DDPM | 68.88 | 6.96 ± 0.10 |
| 60 | SC-VAE | 86.18 | 3.97 ± 0.05 |

Table 2: Ablation study of the SCDM bottleneck on CIFAR-10 with 30% noisy inputs.

| Method | FID $\downarrow$ | IS $\uparrow$ |
|---|---|---|
| SCDM-P (Projection only) | 24.23 | $8.41 \pm 0.08$ |
| SCDM-I (ISTA only) | 25.90 | $8.85 \pm 0.13$ |
| **Full SCDM** | **23.96** | $\mathbf{8.97 \pm 0.14}$ |

Both SCDM-P and SCDM-I achieve reasonable performance, but each is consistently inferior to the full SCDM module. These results indicate that neither component alone is sufficient, and that the combined design is necessary to obtain the best generation quality.

## 5.2 DATA GENERATION WITH SCDM

In this section, we conduct experiments to study the performance of the SCDM on datasets of physical scenarios. The purpose of our experiment is not only to compete with a large number of generative models, but also to explore the compactness of the latent variable $x_t$ and the performance changes of the model as the number of sparse layers K increases. And we also need to explore whether sparse representations can truly uncover the potential factors in the dataset. To facilitate a clear assessment of model behavior, we first test SCDM on a deliberately simple dataset (trajectory of a circle). Then, we trained on datasets physical scenarios to test the generalization of SCDM.

**Baselines.** To quantify the accuracy, we compared SCDM with four baselines. **SDD** Ostheimer et al. (2025) augments each dimension with a binary sparsity bit that is diffused alongside the data. **DDIM** Song et al. (2020) is a non markovian deterministic variant of DDPM that accelerates sampling. **PIDM** Bastek et al. (2025) augments diffusion with physics-informed residuals to respect governing equations, offering a complementary form of inductive bias. We also considered **DDPM**.

**Datasets** All datasets are from the Supporting Online Material[1] of Schmidt & Lipson (2009). The datasets were recorded on standard tabletop mechanics setups (air track oscillators and single and double pendulum rigs) using video tracking. The SOM provides frame-by-frame trajectories; velocities and accelerations are obtained by numerically differentiating the recorded position/angle traces (after standard smoothing), as detailed in the SOM.

We use one synthetic dataset consisting of 360 equally spaced 2-D points sampled from the circle $x^2 + y^2 = 16$ (variables x,y). In addition, we use six motion-tracked trajectory datasets. They are: (1) double linear oscillator (DL), $x_1, x_2, v_1, v_2$, 495 points; (2) single linear oscillator with acceleration (LA), x,v,a, 870 points; (3) single linear oscillator (LH), x,v, 870 points; (4) pendulum in angular coordinates (PA), $\theta, \omega, \alpha$, 556 points; (5) pendulum in Cartesian coordinates (PC), x,y, 600 points; (6) pendulum in angular coordinates without acceleration (PH), $\theta, \omega$, 556 points.

The data is normalized by zero mean and unit variance and then input into SCDM: The diffusion process adopts the cosine $\beta$-schedule ($T = 1000$, $\beta_1 \approx 10^{-5}$ to $\beta_T \approx 0.02$), and Gaussian noise is gradually injected. The compression process follows equations 14 and 16: subspace projection yields a compressed representation, which is subsequently sparsified via ISTA. Notably, the choice of threshold $\lambda$ has little impact on model performance (see Appendix). We fix $\lambda$ to 0.5 in all subsequent experiments. The Fourier embedding of timestep $t$ is concatenated with $z_t$ and passed through a three-layer GELU-MLP decoder to predict the noise component. AdamW was adopted in training ($lr = 1e-3$, $wd = 1e-4$), and EMA (decay = 0.995) was introduced to enhance the stability.

## 5.3 EVALUATION OF MODEL EFFECTIVENESS

To verify the correctness of the generated data, we use three established metrics: average radius absolute error ($R_{\mathrm{MAE}}$), center offset percentage (MDN%CE), and area error percentage (% VFE). To maximize the quality of generated samples, we incorporate both the original latent representation $z_t$ and its sparse correction $z_{t-1}$, enabling the decoder to exploit complementary global and structured information.

---

[1]http://www.sciencemag.org/cgi/content/full/324/5923/81/DC1

Table 3: Sampling performance across seven motion datasets. CT: circle; DL: double linear oscillator; LA: single linear oscillator; PA: pendulum–angular; PC: pendulum–Cartesian; LH: low-dimensional oscillator; PH: low-dimensional pendulum.

| Metric | Model | CT | DL | LA | PA | PC | LH | PH |
|---|---|---|---|---|---|---|---|---|
| RMAE (%) | SCDM | **1.37** | 20.04 | **24.59** | **35.00** | 49.44 | **10.17** | **27.41** |
| | DDPM | 4.00 | 19.99 | 26.67 | 35.21 | 47.38 | 10.29 | 28.05 |
| | DDIM | 4.01 | 20.09 | 33.02 | 49.12 | 62.19 | 11.06 | 31.41 |
| | SDD | 2.67 | **19.64** | 26.06 | 72.85 | **43.78** | 10.44 | 27.65 |
| | PIDM | 2.27 | 54.69 | 52.23 | 77.40 | 64.16 | 11.34 | 88.9 |
| MDN %CE | SCDM | 4.14 | **5.58** | **6.80** | 3.78 | **3.49** | **0.94** | **2.67** |
| | DDPM | 7.21 | 7.09 | 8.61 | 3.47 | 6.97 | 7.57 | 3.64 |
| | DDIM | 18.15 | 38.58 | 33.76 | 17.95 | 40.87 | 31.16 | 37.86 |
| | SDD | 4.76 | 12.57 | 14.63 | **2.49** | 15.08 | 5.84 | 4.44 |
| | PIDM | 11.13 | 81.26 | 10.46 | 13.95 | 3.71 | 10.08 | 17.1 |
| %VFE | SCDM | **+0.05** | **+0.07** | +1.64 | +0.43 | **+1.51** | +1.21 | +1.29 |
| | DDPM | +0.39 | **+0.07** | **-1.09** | **+0.41** | +8.38 | **+0.49** | -0.65 |
| | DDIM | +0.34 | +1.08 | +9.98 | +28.36 | +54.77 | +6.23 | +22.12 |
| | SDD | -0.05 | -1.14 | -1.25 | -1.85 | +8.44 | +1.59 | +1.68 |
| | PIDM | +2.74 | -5.66 | -6.17 | -17.17 | -36.24 | -6.11 | **-0.23** |

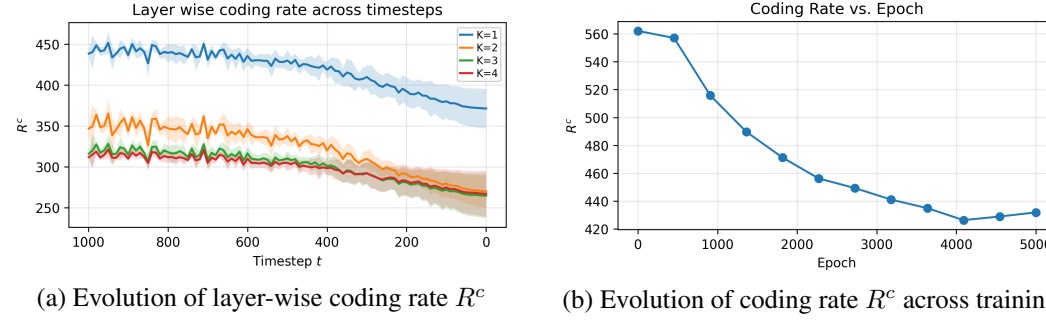

(a) Evolution of layer-wise coding rate $R^c$ across diffusion timesteps

(b) Evolution of coding rate $R^c$ across training epochs

Figure 3: Comparison of coding rate $R^c$ dynamics: (a) layer-wise across diffusion timesteps; (b) evolution across training epochs.

We evaluate our SCDM model on several real-world physical systems, including linear oscillators and pendulums tracked via motion capture. Because the original image-centric backbones are not directly applicable to our low-dimensional dynamics, we use a shared MLP backbone across methods. SCDM augments it with a learned ISTA bottleneck (K=5) while DDPM uses the plain backbone. DDIM denotes a deterministic sampling scheme applied to the DDPM-trained weights. We uniformly use the energy conservation formula as residuals for all datasets except the circular dataset. All other settings (diffusion schedule, optimizer, normalization, and data splits) are kept identical.

As shown in Table 3, SCDM achieves consistently strong performance across seven distinct real-world motion datasets, significantly outperforming baseline methods across multiple evaluation metrics. On challenging systems such as the nonlinear pendulum (PA) and Cartesian trajectories (PC), SCDM further exhibits superior compactness and plausibility compared to baseline models. These results collectively validate that SCDM not only preserves fidelity across simple tasks but also generalizes robustly to complex physical dynamics. We visualized the results of SCDM in the Appendix (5).

## 5.4 CODING RATE EVOLUTION

To probe how the sparse stack influences representation compression over time, we track the layer-wise coding rate $R^c$ across diffusion timesteps in an SCDM configured with a sparsity depth of K=7. Figure 3 (a) plots the first four layers (K=1 to 4) for clarity. The curves reveal that the coding rate

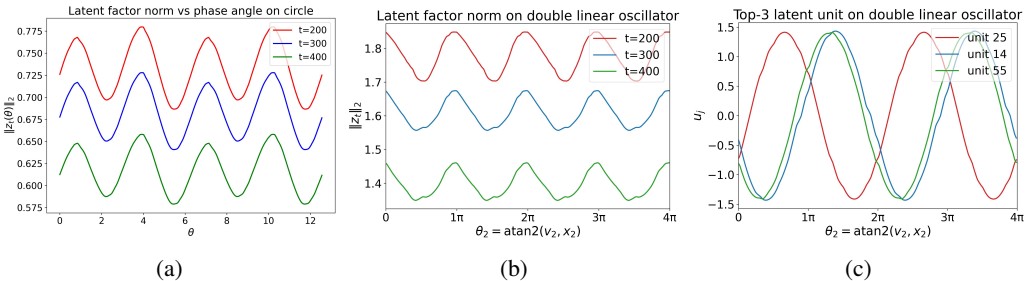

(a)                                     (b)                                     (c)

Figure 4: Latent representation analysis on circle and double linear oscillator datasets. (a) Latent factor norm on the circle dataset at different reverse steps ($t = 200/300/400$). (b) Latent factor norm on the double linear oscillator dataset at different reverse steps ($t = 200/300/400$). (c) The first three latent dimensions (with highest variance) on the double linear oscillator dataset, illustrating the dominant components of the learned representation.

drops sharply up to the third layer, after which it plateaus, indicating that most of the achievable compression is already realized by K=3; additional layers yield only marginal gains in sparsity.

The epoch-wise analysis of the coding rate ($R^c$), shown in Figure 3 (b), further provides insights into the training dynamics of the SCDM. The monotonic decline of $R^c$ across epochs indicates progressive refinement and stabilization of the sparse latent representations. After a steep initial decrease, the rate stabilizes after approximately 3,000 epochs, signaling convergence toward a minimal yet sufficient representation complexity. This behavior underscores that the learned sparse representation effectively captures essential structural information, facilitating efficient and stable generative performance.

### 5.5 LATENT FACTOR ANALYSIS

To probe what SCDM learns, we adopt a unified setup: an SCDM with K=5 sparse-coding steps trained for 5000 epochs under the standard objective. After training, we freeze all parameters (including step-dependent representation weights) and read out the intermediate sparse codes at fixed reverse steps $t \in \{200, 300, 400\}$ for analysis. We deliberately move from simple to complex periodic motion: on the circle dataset where inputs are $(x, y) = (\cos\theta, \sin\theta)$, we sweep synthetic $\theta \in [0, 4\pi]$. For each $\theta$, we apply the same normalization and diffusion scaling used during training to obtain $\mathbf{x}_t$, and compute the final sparse code $z_t(\theta)$. We then visualize the curve $\|z_t(\theta)\|_2$ versus $\theta$. As shown in Figure 4 (a), during the reverse denoising process (as t decreases), the sinusoidal structure gradually emerges with larger amplitude, indicating that the model reconstructs a periodic latent aligned with the data geometry. On the double linear oscillator with state $(x_1, v_1, x_2, v_2)$ (each pair in harmonic motion, $v_i = \dot{x}_i$), the aggregated norm mixes contributions from both oscillators; we therefore fix $(x_1, v_1)$ and sweep the second oscillator's phase $\theta_2 = \mathrm{atan2}(v_2, x_2)\, using\, x_2(\theta_2) = \cos\theta_2,\, v_2(\theta_2) = -\sin\theta_2$. The resulting $\|z_t(\theta_2)\|_2$ behaves like a superposition of a few sinusoids (Fig.4 (b)). To disentangle this mixture, we inspect the three highest-variance latent coordinates $u_j(\theta_2)$. Each traces a near-sinusoid with distinct phase and weight, confirming that SCDM encodes the dynamics via a small set of trigonometric components consistent with the underlying factors (Fig. 4 (c)).

## 6 CONCLUSION

In this paper, we introduce SCDM, a novel framework integrating sparse rate reduction with diffusion modeling to enforce structured sparsity throughout the reverse compression process. By leveraging explicit sparse constraints, SCDM achieves robust and latent representations, significantly stabilizing the generative process and closely matching underlying latent factors. Empirical results across CIFAR-10 and several datasets of physical scenarios demonstrate SCDM's superior performance and generalization capabilities, confirming that embedding structured sparsity enhances generative accuracy.

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

# A APPENDIX

# B COMPUTATIONAL INFRASTRUCTURE FOR EXPERIMENTS

Experiments ran on a single NVIDIA RTX A6000 (48 GB, Driver 535.230.02, CUDA 12.2) under Ubuntu 22.04. Code was executed in PyTorch 2.3.

# C IMPACT OF HYPERPARAMETERS ON MODEL PERFORMANCE

We conduct a sensitivity study on the initial shrinkage parameter $\lambda$ under a fixed sparsity depth $K = 5$. As summarized in Table 4, the reconstruction accuracy ($R_{\mathrm{MAE}}$), sparsity level (MDN % CE), and physical consistency (% VFE) remain relatively stable across a wide range of $\lambda$ values (from 0.01 to 0.9). This indicates that SCDM is robust to the choice of $\lambda$, and does not require extensive tuning for effective performance.

Table 4: Effect of sparsity threshold $\lambda$ on model performance ($K = 5$).

| $\lambda$ | $R_{\mathrm{MAE}}$ | MDN % CE | % VFE |
|---|---|---|---|
| 0.01 | 1.662% | 5.02% | 0.60% |
| 0.10 | 1.515% | 5.12% | 0.30% |
| 0.50 | 1.403% | 4.75% | 0.16% |
| 0.70 | 1.444% | 5.07% | 0.07% |
| 0.90 | 1.908% | 4.67% | -0.10% |
| 1.00 | 1.973% | 4.80% | -0.07% |

Furthermore, we explored the effect of sparsity depth $K$ on sampling performance, as detailed in Table 5. Moderate sparsity depths provide slight improvements in model accuracy compared to the single-layer configuration ($K = 1$), indicating a subtle but consistent benefit from increased sparsity stacking. As $K$ grows beyond 4, the metrics exhibit minor fluctuations rather than significant degradation, reflecting overall stable performance and robustness across varying sparsity depths. This suggests that the model achieves stable performance regardless of deeper sparse architectures.

Table 5: Evaluation of SCDM sampling performance at different sparse layers $K$.

| $K$ | $R_{\mathrm{MAE}}$ | MDN % CE | % VFE |
|---|---|---|---|
| 1 | 2.28% | 4.76% | $-0.08\%$ |
| 2 | 2.12% | 4.67% | $-0.08\%$ |
| 3 | 2.13% | 4.27% | $+0.31\%$ |
| 4 | 2.09% | 4.18% | $+0.19\%$ |
| 5 | 2.14% | 4.27% | $+0.21\%$ |
| 6 | 2.15% | 4.06% | $+0.22\%$ |
| 7 | 2.10% | 4.17% | $+0.26\%$ |

## D  REAL PHYSICAL SYSTEMS

To better demonstrate the generalization performance of SCDM, in this section, we have visualized the first two dimensions of the SCDM sampled in the real world.

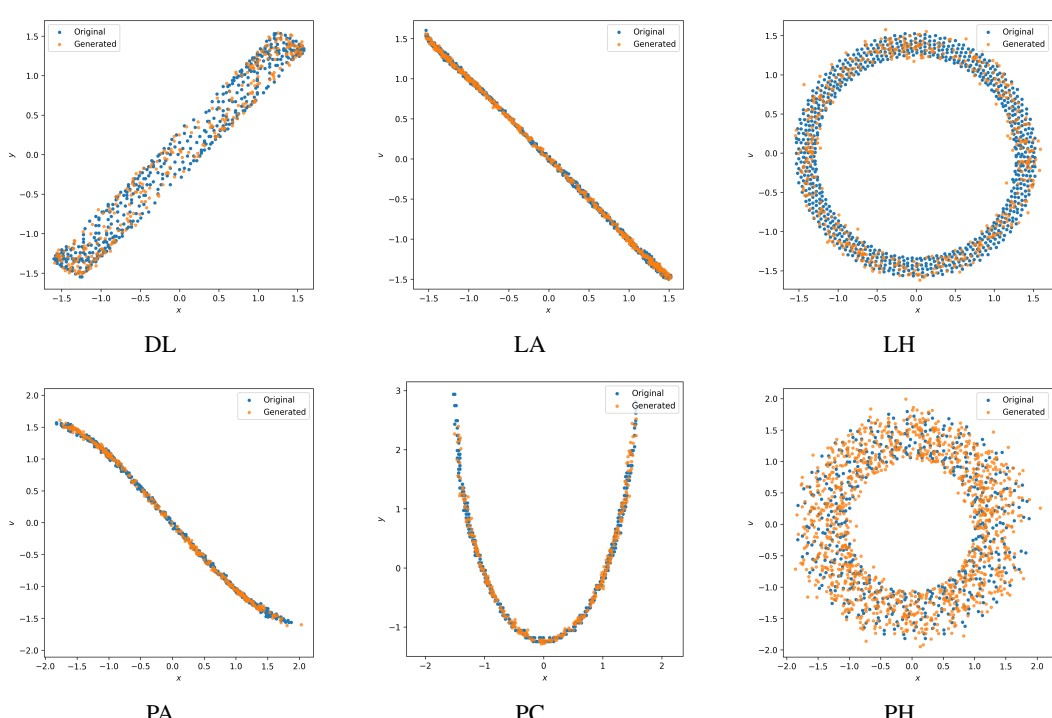

Figure 5: Visualization of the six datasets: DL (double linear oscillator, $x_1, x_2, v_1, v_2$); LA (single linear oscillator, $x, v, a$); LH (low-dimensional linear oscillator); PA (pendulum in angular coords, $\theta, \omega, \alpha$); PC (pendulum in Cartesian coords, $x, y$); PH (low-dimensional pendulum).

Figure 5 demonstrates that SCDM accurately recreates the phase portraits of all six datasets. In DL, synthetic points overlap the tilted rectangular orbit, showing the model captures coupled linear relations. For LA, generated data align with the $x$–$v$ line, matching the deterministic dynamics. LH's concentric ring is reproduced with uniform radius, indicating preserved periodicity. PA's curved $\theta$–$\omega$ trajectory and variance are recovered, evidencing robustness to nonlinear dynamics. In PC,

the parabolic locus in (x,y) space is retained, confirming coordinate invariance. PH's annular distribution appears with similar radial spread and density. These results confirm SCDM's ability to generalize across heterogeneous physical systems.

# E CIFAR10

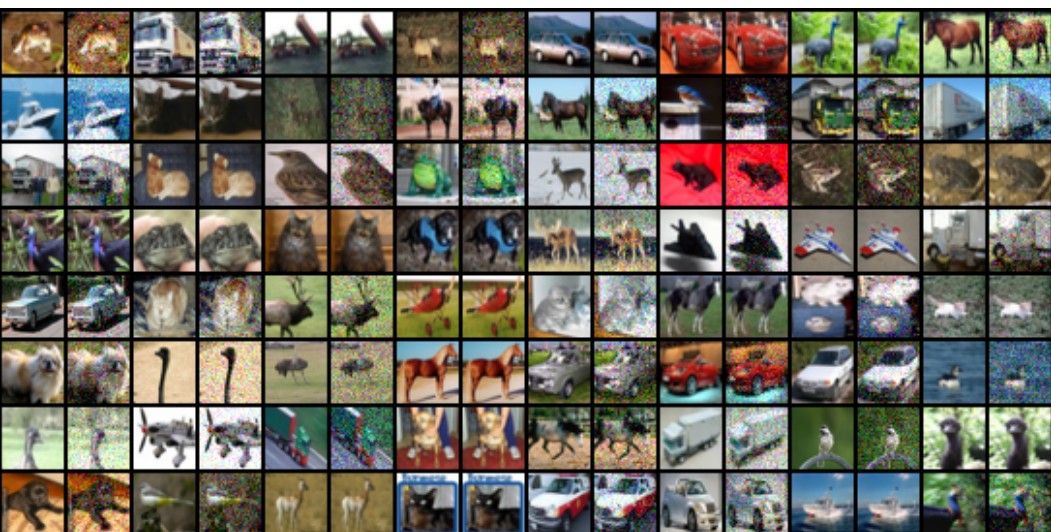

Figure 6: Visualization of the training data under 50% noise corruption on CIFAR10. Odd columns show clean training images, while even columns show their counterparts after adding Gaussian noise with 50% corruption ratio.

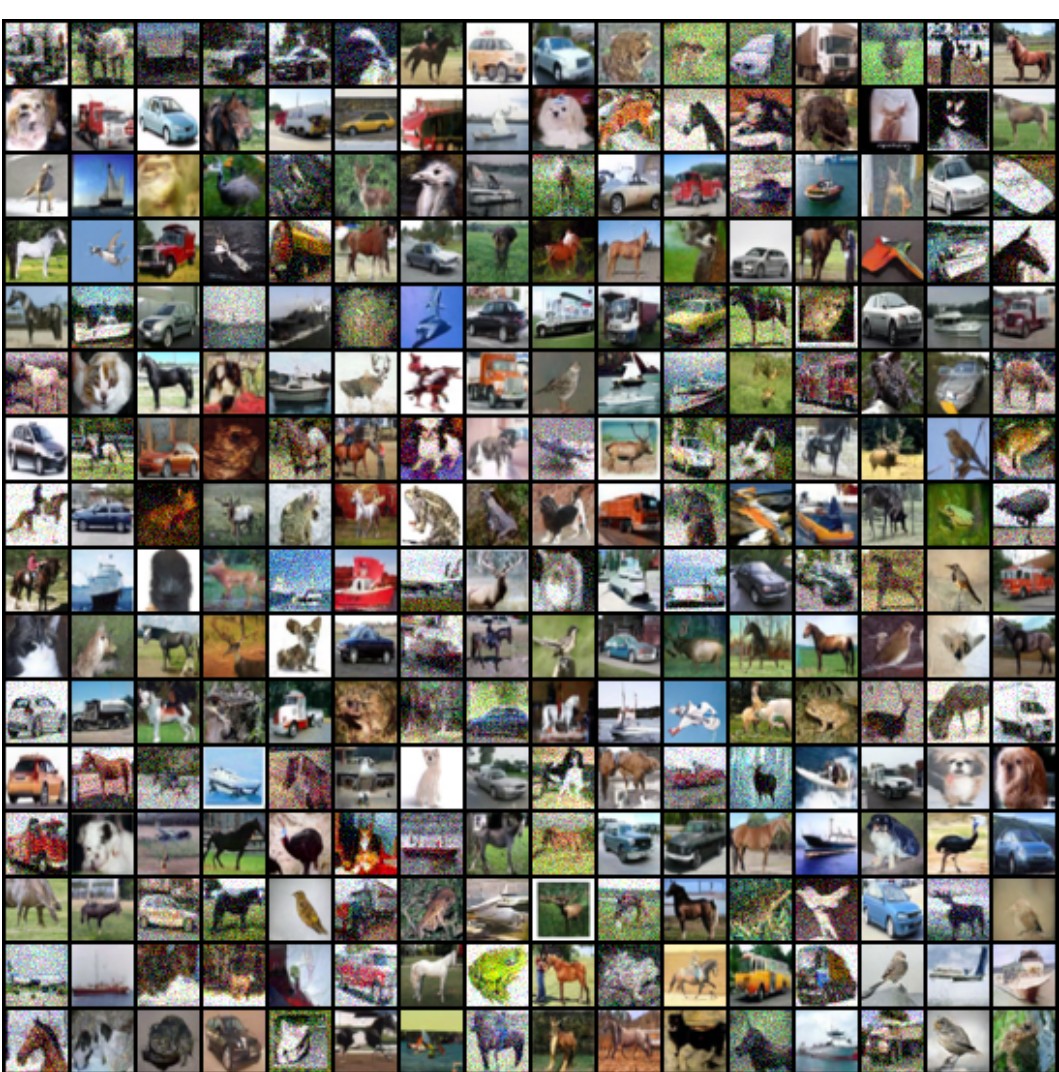

Figure 7: CIFAR-10 samples generated by SCDM trained on the 30% noisy training set. FID=23.96.

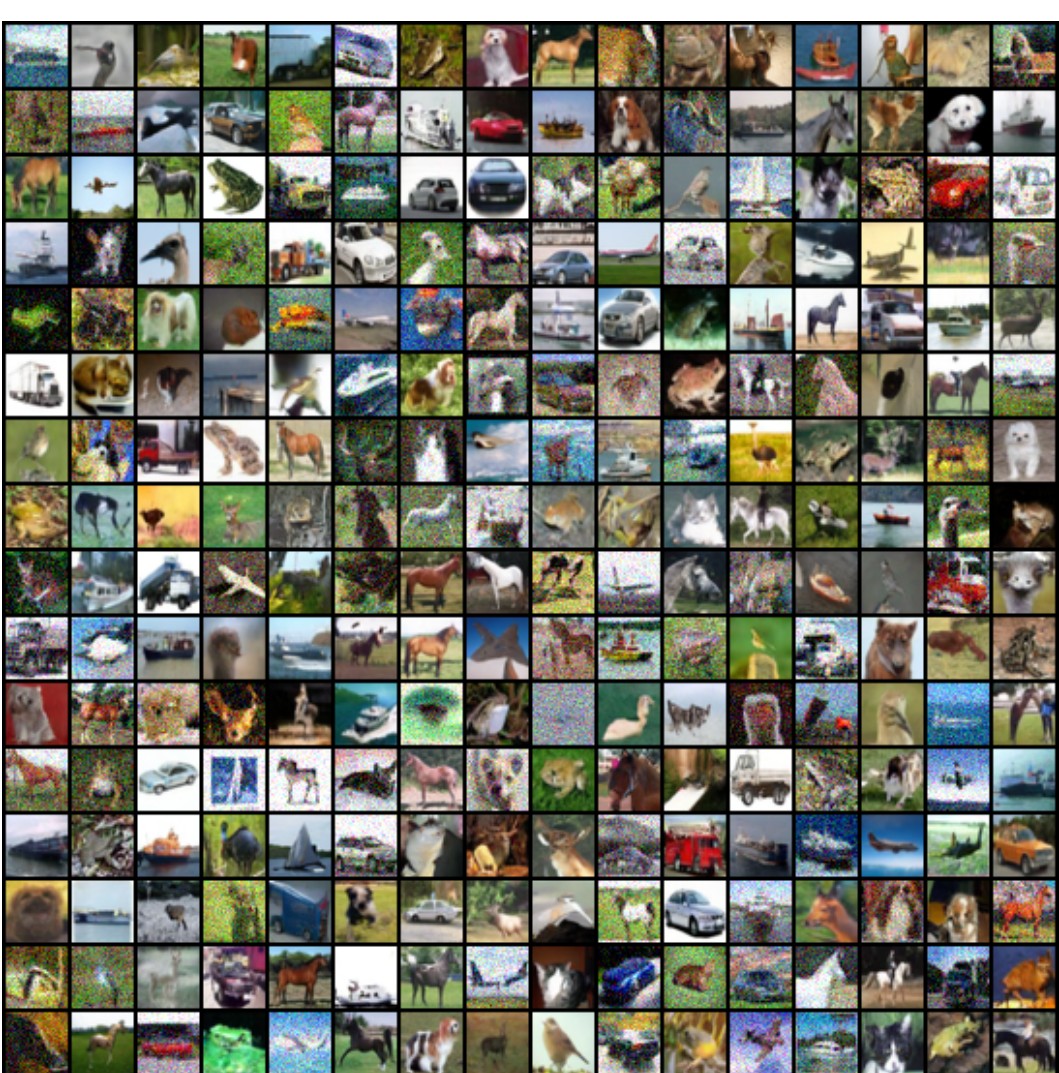

Figure 8: CIFAR-10 samples generated by SCDM trained on the 40% noisy training set. FID=35.561.

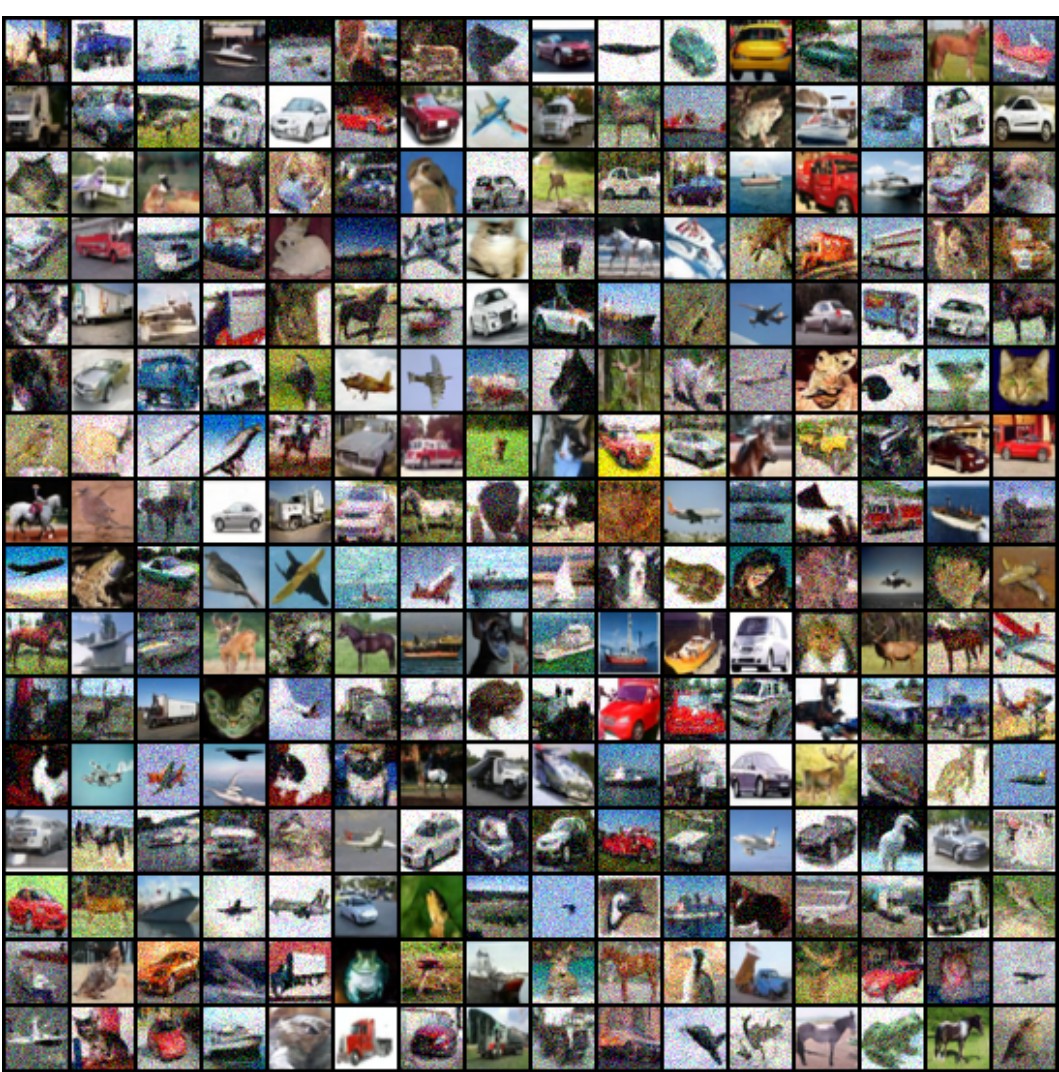

Figure 9: CIFAR-10 samples generated by SCDM trained on the 50% noisy training set. FID=53.069.

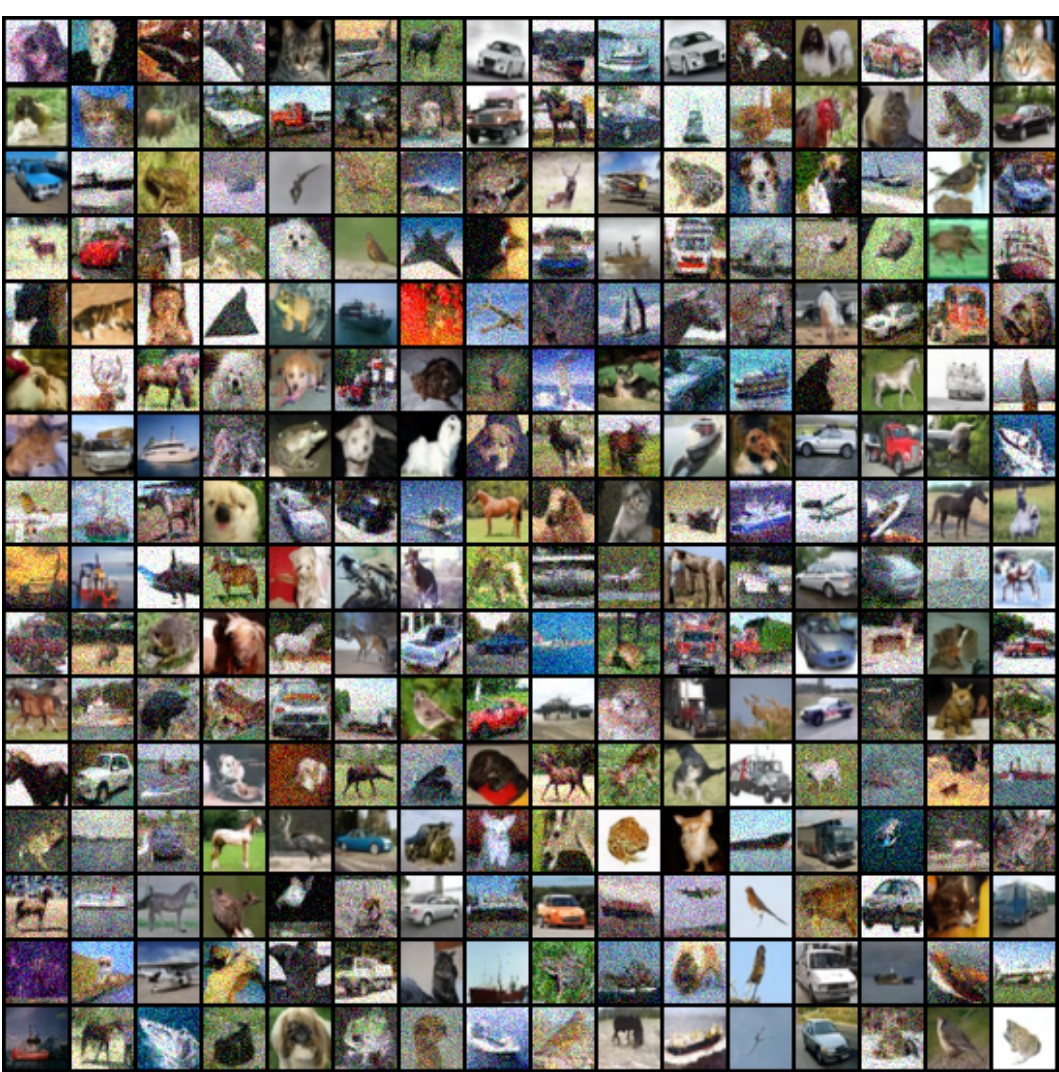

Figure 10: CIFAR-10 samples generated by SCDM trained on the 60% noisy training set. FID=61.088.

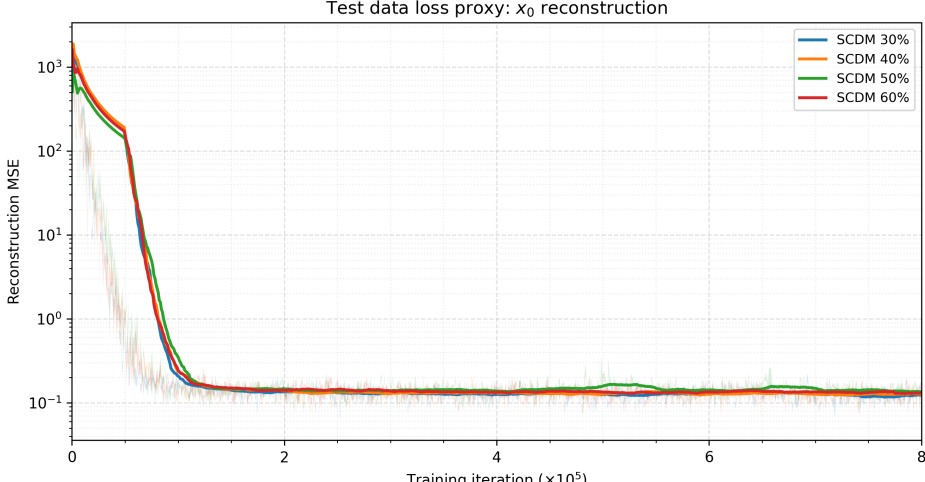

Figure 11: Test data loss proxy on CIFAR-10: EMA-smoothed $x_0$ reconstruction MSE under different noisy training ratios (30%, 40%, 50%, 60%).

