# OpenReview forum: "Sparse-Compression Diffusion Models"
_ICLR.cc/2026/Conference — Submitted to ICLR 2026_

### Official Review · Reviewer_g5Ep · 2025-10-27

**Soundness:** 1
**Presentation:** 1
**Contribution:** 2
**Rating:** 0
**Confidence:** 5

**Summary:**

The paper proposes SCDM, a diffusion framework that inserts a compression-sparsity module in each reverse step: encode $x_t\rightarrow z_t$, project onto a mixture of orthogonal subspaces, apply an ISTA-style soft-threshold, and then predict the noise from the resulting sparse code to form the usual DDPM update rule. The claimed contributions are (1) integrating "sparse rate reduction" into diffusion, and (2) demonstrating that sparse latents "align with scientific theories," using toy/small physics datasets (circle, oscillators, pendulums).

**Strengths:**

Pursuing a sparse latent structure during reverse diffusion is a reasonable direction. The paper attempts to formalize this via rate-reduction terms (eq. 5–8).

**Weaknesses:**

1. The problem statement is very unclear and overstated. The paper asserts that standard reverse diffusion "merely overfits in data denoising" and "obstructs… robust physical synthetic data," but provides no evidence or references establishing this problem as real or general. This is presented in the abstract/introduction and is used to justify the method. However, the experiments do not demonstrate real-world failures of DDPM beyond small toy settings, and does not reveal why is it that DDPM fails. To me, this appears to be a flaw in the experiments.
2. The core claim that the learned latents "align well with scientific theories" is asserted (what does scientific theories even mean?), but the evidence is qualitative plots of sinusoidal-like activation norms (Fig. 4) and small metric gains on tiny datasets. This does not validate alignment with the governing physical laws in a robust way.
3. The projection operator in algorithm 1 is not defined clearly anywhere in the paper.
4. Eq. 14 mixes the usual DDPM mean with $z_{t-1}^{K}$ in place of $x_{t}$. There is no derivation showing that this substitution yields a valid reverse process.
5. All datasets are tiny and insufficient to evaluate a general generative modeling claim about "robust physical data synthesis."
6. The motivation and what exactly is new are very hard to track.
7. Numerous grammatical errors and awkward phrasing (e.g., "pictural," "merely overfits in data denoising" in the abstract), plus inconsistent use of technical terms, make the paper feel unpolished.

**Questions:**

See weaknesses.

---

> ### Author Response · Authors · 2025-11-23
>
> Dear Reviewer g5Ep:
>
> Thank you for the detailed and constructive feedbacks on our submission.  Below, we address your concerns and outline our responses to your feedbacks.
>
> W1 (The problem statement):
>
> In the new revised version, we have made corrections and unified definitions in response to your suggestions.
>
> We also note that using high-level language to describe internal representations is common in the representation-learning literature.
> For instance, Yu et al. (2023) [1] and Buchanan et al. (2025) [2] both discuss how learned features capture “intrinsic structure” or “underlying factors” of data distributions.
>
> W2 (Experiment):
>
> To further explore the failures of DDPM, we have added an experiment on CIFAR10 with noisy training data (Section 5.1 in the revised manuscript).  Specifically, we corrupt the 50k training images with different levels of additive Gaussian noise (Noise = 0%, 30%, 40%, 50%, 60%), while keeping the 10k test images clean. We then train DDPM, SC-VAE, and our SCDM under identical settings and evaluate standard image generation metrics, i.e., FID and Inception Score (IS), as summarized in Table 1. SCDM delivers great performance across all settings, and its advantage becomes even more pronounced when noise is injected into the training data. We also provide, in the Appendix E, qualitative samples generated under various noise levels and the associated training loss curves.
>
> We conduct ablations on the 30% noisy CIFAR-10 dataset by isolating (i) SCDM-P : the low-dimensional projection in Eq 13 and (ii) SCDM-I : the ISTA-style sparsification in Eq 15. Both SCDM-P and SCDM-I achieve reasonable performance, but each is consistently inferior to the full SCDM module. The result indicates that neither component alone is sufficient, and that the combined design is necessary to obtain the best generation quality.
>
> In section 5.2, we newly add basleine PIDM to enhance the completeness of the experiment. We uniformly use the energy conservation formula as residuals for all datasets except the circular dataset.
>
> W3(The motivation):
>
> We have restated the theoretical part in Section 4.1 MODEL FRAMEWORK, and the updated content is highlighted in blue. Specifically, from the forward process, we can have $\mathbb{E}[\varepsilon \mid x_t]
> = \dfrac{1}{\sqrt{1-\bar{\alpha}_t}}
> \Big( x_t - \sqrt{\bar{\alpha}_t}\,\mathbb{E}[x_0 \mid x_t] \Big)$.
>
> This makes explicit that $\widehat{\varepsilon_\theta}\left(x_t,t\right)$ is defined conditionally on $x_t$, and that the noise prediction network is in fact learning a function that is highly correlated with the posterior mean $\mathbb{E}[x_0\mid x_t]$.
>
> Our goal in SCDM is precisely to learn a sparse representation $z_t$ that preserves the most essential factors of variation in $x_t$, and to use these factors to predict $\widehat{\varepsilon_\theta}\left(z_t,t\right)$. To the best of our knowledge, explicitly inserting a learned representation bottleneck on $x_t$ at each diffusion step and training the model to predict the noise from this representation $z_t$ has not been explored in prior diffusion work. For the representation objective, we adopt the CRATE-style coding rate formulation as a concrete and effective choice. In the same spirit as Xiao et al.(2025) [3], which introduces a new VAE variant that integrates a known sparse-coding module into the VAE framework. Analogously, our contribution lies in integrating a structured sparse representation module into the diffusion reverse process.
>
> Please re-evaluate our revised manuscript.
>
> Best regards,
>
> ICLR 2026 Conference Submission6110 Authors
>
> --------------------------------------------
> [1] Yaodong Yu, Sam Buchanan, Druv Pai, Tianzhe Chu, Ziyang Wu, Shengbang Tong, Benjamin Haeffele, and Yi Ma. White-box transformers via sparse rate reduction. Advances in Neural Information Processing Systems, 36:9422–9457, 2023.
>
> [2] Sam Buchanan, Druv Pai, Peng Wang, and Yi Ma. Learning Deep Representations of Data Distributions. Online, August 2025. https://ma-lab-berkeley.github.io/deep-representation-learning-book/.
>
> [3] Pan Xiao, Peijie Qiu, Sung Min Ha, Abdalla Bani, Shuang Zhou, and Aristeidis Sotiras. Sc-vae:Sparse coding-based variational autoencoder with learned ista. Pattern Recognition, 161:111187,2025.

---

> > ### Comment · Reviewer_g5Ep · 2025-11-25
> > **Response to rebuttal**
> >
> > I thank the authors for their response.
> >
> > __W1:__ Can you please highlight for me exactly what was changed? I don't see any change in the abstract, introduction, etc., where you introduce your work and explain what problem you are trying to solve. Of course, it is definitely okay to use high-level language, but in this case, I just don't understand what's going on. Here are a few constructive examples, some of which I stated in my original review: In the abstract, please explain to me exactly what you mean by
> > 1. "the conventional reverse diffusion process merely overfits in data denoising."
> > 2. "the denoising mechanism obstructs the diffusion models to generate robust physical synthetic data."
> > 3. "a novel reverse diffusion process that enforces compression."
> > 4. "enabling the ability to capture the intrinsic factors of some rules."
> >
> > __W2:__ I am sorry, but I am failing to understand what's the point of these new experiments? Were you trying to achieve good generation performance (e.g., FID)? If so, how does it relate to the problem your paper is trying to solve? (i.e., "capturing the intrinsic factors of some rules").
> >
> > __W3:__ What do you mean by "the most essential factors of variation in $x_t$"?
> >
> > While there may be good ideas in this paper, unfortunately, it is still too difficult to parse.

---

> > > ### Author Response · Authors · 2025-11-26
> > >
> > > Dear Reviewer g5Ep:
> > >
> > > Thank you for the detailed and constructive feedbacks on our submission. Below, we address your concerns and outline our responses to your feedbacks.
> > >
> > > $\textbf{Q1:Motivation of our paper}$
> > >
> > > The distribution of data, especially in physical scenarios, are dominated by latent factor constraints. In this paper, we would like to explore the possibility to train a generative model that is implicitly restricted by the learned latent factors.
> > >
> > > $\textbf{Q2: Methodology of our paper}$
> > >
> > > We assume the noise in $x_t$ is composed of both the noise upon the structural component of $x_0$ and the other noises. In order to obtain improved $x_0$, we use representation learning to extract the principal structural directions of $x_t$ to guide the generation of $x_{t-1}$.
> > >
> > > In our implementation, we use an NN $T$ to encoder $x_t$ into a unified representation space $z_t$ refer to algorithm line 14. In order to extract the principal structural directions of $x_t$, we apply sparse compression in representation space and produce $z_{t-1}^K $ refer to algorithm line 15 to 18.
> > >
> > > In order to obtain $x_{t-1}$, we map $z_{t-1}^K$ to same ambient space as $x_t$ through an NN decoder $T^{-1}$. And we refer to the standard sample process (algorithm 19 to 22)
> > >
> > > $\mu_\theta\left(z_{t-1}^K,t\right)=\frac{1}{\sqrt{\alpha_t}}\left(T^{-1}\left(z_{t-1}^K\right)-\frac{\beta_t}{\sqrt{1-\bar{\alpha_t}}}g_\theta\left(T^{-1}\left(z_{t-1}^K\right),t\right)\right)$
> > >
> > > $x_{t-1}=\mu_\theta\left(z_{t-1}^K,t\right)+\sigma_t\gamma$
> > >
> > > where $\gamma\sim\mathcal{N}\left(0,I\right)\mathrm{\ \ if\ }t>1,\ \mathrm{else\ \gamma}=0$. Finally, we can obtain the improved $x_0$.
> > >
> > > $\textbf{Q3: The motivation of the Experiment}$
> > >
> > > The assumption that the noise in $x_t$ is composed of both the noise upon the structural component of $x_0$ and the other noises is supported in section 5.1 IMAGE GENERATION WITH SCDM. If the initial data $x_0$ is contaminated, then $x_t$ actually contains other noises besides the noise upon the structure of $x_0$. We conducted experiments on data contaminated to different degrees and found that this two-step denoising can perform well.
> > >
> > > And we conduct ablations on the 30% noisy CIFAR-10 dataset by isolating (i) the low-dimensional projection in Eq 14 and (ii) the ISTA-style sparsification in Eq 16. The result indicates that neither component alone is sufficient, and that the combined design is necessary to obtain the best generation quality. The result indicates that neither component alone is sufficient, and that the combined design is necessary to obtain the best generation quality.
> > >
> > > In section (5.2-5.5), we extended SCDM to seven physical datasets to evaluate the generation quality, examined the impact of sparsity on the generation quality, and we attempted to explore the latent factor learned by SCDM for x_t.
> > >
> > > $\textbf{Q4: Meaning of most essential factors}$
> > >
> > > The essential factors are the structural components of $x_0$ in $x_t$. It is the latent factor SCDM learned. The various expressions of latent factor we used in the paper might have caused you some confusion. We have unified them all as latent factor.
> > >
> > > Please re-evaluate our revised manuscript.
> > >
> > > Best rewards,
> > >
> > >
> > > ICLR 2026 Conference Submission6110 Authors

---

### Official Review · Reviewer_otdE · 2025-10-30

**Soundness:** 1
**Presentation:** 3
**Contribution:** 1
**Rating:** 2
**Confidence:** 4

**Summary:**

The paper proposes Sparse-Compression Diffusion Models (SCDM), an extension of DDPMs that introduces a compression–sparsity mechanism into the reverse diffusion step. The method draws heavily on the CRATE (Yu et al., 2023) framework for sparse rate reduction, combining subspace projection and ISTA-style soft-thresholding to encourage low-dimensional latent representations. The authors claim this yields more interpretable and physically robust generative models, supported by experiments on simple physical datasets (oscillators and pendula).

**Strengths:**

1. The idea of embedding sparse representation principles into diffusion modeling is conceptually sound and moderately novel in motivation.
2. The paper is structurally clear and provides a full algorithmic description (Algorithm 1), which makes replication possible in principle.
3. The attempt to link sparse representations with physical interpretability is an appealing research direction.

**Weaknesses:**

1. Limited Theoretical Novelty
- Nearly all equations—including the compression objective $$R(z)-R_c(z;U)-\lambda\|z\|_0$$, the log-det rate formulation, the softmax-weighted subspace projection, and the ISTA sparsification step—are taken directly from the CRATE paper with minimal adaptation.
- The only  change is placing these CRATE blocks inside the DDPM reverse step and feeding the sparse code into the noise predictor. This is an implementation variant, not a new theory.
- There is no probabilistic or variational derivation showing that the added term improves the diffusion likelihood or score estimation.
2. Weak Mathematical Rigor
- Notation and definitions are inconsistent (e.g., non-differentiable $$ \| \cdot \|_0 $$ with no surrogate use).
- The information-theoretic "coding rate" is adopted from CRATE without re-deriving its relevance to diffusion modeling.
- No analysis is given of how the added compression term affects convergence or stability of the reverse process.
3. Insufficient Empirical Rigor
- Metrics: RMAE, MDN %CE, and %VFE measure geometric fidelity of trajectories, not generative quality. They do not test likelihood, diversity, or coverage.
- Baselines: The comparison excludes the 'Physics-Informed Diffusion Model' (Shu et al., JCP 2023) despite citing it. SCDM also uses an extra ISTA bottleneck, so model capacity differs from DDPM/DDIM baselines.
- Datasets: All datasets are low-dimensional (2–4 vars) and small. These tasks are trivial for diffusion models and cannot demonstrate generalization or robustness.
- Reporting:
    -- No multiple seeds, confidence intervals, or significance tests. Figures lack scales and quantitative analysis.
    -- Not clear how individual CRATE components influence sample quality
4. Citation and Scholarly Issues
- Roughly half the references are arXiv-only. Foundational works (VAE, DDPM, SSC) have wrong years or missing venues(Arxiv references even  for papers which have been published in ML venues)
5. Writing and Presentation
- English usage is often incorrect ("successful generating," "merely overfits in data denoising") and many claims are vague ("intrinsic factors of some rules").
- Assertions of "alignment with scientific theories" are rhetorical; no quantitative mapping to actual equations is shown.

**Questions:**

1. How does the compression–sparsity term modify or complement the diffusion ELBO?
2. Why are standard generative metrics (NLL, MMD, FID) absent?
3. Why was the physics-informed diffusion baseline omitted despite citation?
4. Could simpler non-diffusion models (e.g., sparse VAE) achieve similar performance on these datasets?

SCDM is incremental and under-evidenced. The theoretical framework is a direct transplant of CRATE into diffusion modeling, and the experiments are confined to trivial physical systems with inadequate baselines and metrics. The work lacks the mathematical and empirical depth required for a top-tier venue.

-----
Recommended Improvements
------
1. Provide a formal derivation connecting the compression–sparsity term to diffusion likelihood or score matching.
2. Expand experiments to higher-dimensional datasets and include standard generative metrics.
3. Add physics-informed diffusion and sparse generative models as baselines.
4. Report mean +- std across multiple runs and perform significance tests.
5. Correct reference list and cite archival versions where available.

---

> ### Author Response · Authors · 2025-11-23
>
> Dear Reviewer otdE:
>
> Thank you for the detailed and constructive feedbacks on our submission.  Below, we address your concerns and outline our responses to your feedbacks.
>
> W1-2:
>
> We have restated the theoretical part in Section 4.1 MODEL FRAMEWORK, and the updated content is highlighted in blue. Specifically, from the forward process, we can have $\mathbb{E}[\varepsilon \mid x_t]
> = \dfrac{1}{\sqrt{1-\bar{\alpha}_t}}
> \Big( x_t - \sqrt{\bar{\alpha}_t}\,\mathbb{E}[x_0 \mid x_t] \Big)$.
>
> This makes explicit that $\widehat{\varepsilon_\theta}\left(x_t,t\right)$ is defined conditionally on $x_t$, and that the noise prediction network is in fact learning a function that is highly correlated with the posterior mean $\mathbb{E}[x_0\mid x_t]$.
>
> Our goal in SCDM is precisely to learn a sparse representation $z_t$ that preserves the most essential factors of variation in $x_t$, and to use these factors to predict $\widehat{\varepsilon_\theta}\left(z_t,t\right)$. To the best of our knowledge, explicitly inserting a learned representation bottleneck on $x_t$ at each diffusion step and training the model to predict the noise from this representation $z_t$ has not been explored in prior diffusion work. For the representation objective, we adopt the CRATE-style coding rate formulation as a concrete and effective choice. In the same spirit as Xiao et al.(2025) [1], which introduces a new VAE variant that integrates a known sparse-coding module into the VAE framework. Analogously, our contribution lies in integrating a structured sparse representation module into the diffusion reverse process.
>
> W3:
>
> We have added an experiment on CIFAR10 with noisy training data (Section 5.1 in the revised manuscript).  Specifically, we corrupt the 50k training images with different levels of additive Gaussian noise (Noise = 0%, 30%, 40%, 50%, 60%), while keeping the 10k test images clean. We then train DDPM, SC-VAE, and our SCDM under identical settings and evaluate standard image generation metrics, i.e., FID and Inception Score (IS), as summarized in Table 1. SCDM delivers great performance across all settings, and its advantage becomes even more pronounced when noise is injected into the training data. We also provide, in the Appendix E, qualitative samples generated under various noise levels and the associated training loss curves.
>
> In the revision, we conduct ablations on the 30% noisy CIFAR-10 dataset by isolating (i) the low-dimensional projection in Eq 13 and (ii) the ISTA-style sparsification in Eq 15. Both SCDM-P and SCDM-I achieve reasonable performance, but each is consistently inferior to the full SCDM module. The result indicates that neither component alone is sufficient, and that the combined design is necessary to obtain the best generation quality.
>
> In section 5.2, we newly added basleine PIDM. We uniformly
> use the energy conservation formula as residuals for all datasets except the circular dataset.
>
> W4-5:
>
> In the new revised version, we have made corrections and unified definitions in response to your suggestions.
>
> We also note that using high-level language to describe internal representations is common in the representation-learning literature.
> For instance, Yu et al. (2023) [2] and Buchanan et al. (2025) [3] both discuss how learned features capture “intrinsic structure” or “underlying factors” of data distributions.
>
>
> Please re-evaluate our revised manuscript.
>
> Best regards,
>
> ICLR 2026 Conference Submission6110 Authors
>
> ----------------------------------------------------------
> [1] Pan Xiao, Peijie Qiu, Sung Min Ha, Abdalla Bani, Shuang Zhou, and Aristeidis Sotiras. Sc-vae:Sparse coding-based variational autoencoder with learned ista. Pattern Recognition, 161:111187, 2025.
>
> [2] Yaodong Yu, Sam Buchanan, Druv Pai, Tianzhe Chu, Ziyang Wu, Shengbang Tong, Benjamin Haeffele, and Yi Ma. White-box transformers via sparse rate reduction. Advances in Neural Information Processing Systems, 36:9422–9457, 2023.
>
> [3] Sam Buchanan, Druv Pai, Peng Wang, and Yi Ma. Learning Deep Representations of Data Distributions. Online, August 2025. https://ma-lab-berkeley.github.io/deep-representation-learning-book/.

---

> ### Comment · Reviewer_otdE · 2025-11-27
> **Response to rebuttal**
>
> I thank the authors for the response and the corrections. but I still have some questions
>
> **W1-2**:
> - I still don't understand the motivation behind not using the Gaussian noise. The statement regarding overfitting in the reverse diffusion process is unclear and possibly incorrect( Line 010-012). Overfitting is a training property, not an inference behavior.  Please clarify whether you mean that the process produces outputs too similar to the training data or that it fails to generalize beyond denoising behavior.
> - If you change the objective as you do in Eq. 13(from the $L_0$-norm penaly to $L_1$-norm term), at least one would expect the results in experiments to change too. The L0-norm counts the number of non-zero elements in your representation. This is not the same as L1-norm. How does this new norm affect the results?
>
> **W3:**
> - can the authors clarify how these energy conservation formulas for the  residuals were derived? this is mentioned neither in the main text nor in the appendix.
> - can you also explain the key message of the added experiment?
>
>
> **W4-5:** As noted in my comment, "intrinsic factors of some rules" is very vague and does not carry any new information. This is not  high language.
>
> Best Regards,
>
> Reviewer otdE

---

> > ### Author Response · Authors · 2025-11-28
> >
> > Dear Reviewer otdE:
> >
> > Thank you for your reply.
> >
> > $\textbf{Motivation of our paper:}$
> >
> >
> > The distribution of data, especially in physical scenarios, are dominated by latent factor constraints. In this paper, we would like to explore the possibility to train a generative model that is implicitly restricted by the learned latent factors.
> >
> >
> > $\textbf{Methodology of our paper:}$
> >
> >
> > We assume the noise in $x_t$ is composed of both the noise upon the structural component of $x_0$ and the other noises. In order to obtain $x_0$, we use representation learning to extract the principal structural directions of $x_t$ to guide the generation of $x_{t-1}$.
> >
> >
> > In our implementation, we use an NN $T$ to encoder $x_t$ to a unified representation space $z_t$ refer to algorithm line 14. In order to extract the principal structural directions of $x_t$, we apply sparse compression in representation space and produce $z_{t-1}^K$ refer to algorithm line 15 to 18.
> >
> >
> > In order to obtain $x_{t-1}$, we map $z_{t-1}^K$ to same ambient space as $x_t$ through an NN decoder $T^{-1}$. And we refer to the standard sample process (algorithm 19 to 22) :
> >
> > $\mu_\theta\left(z_{t-1}^K,t\right)=\frac{1}{\sqrt{\alpha_t}}\left(T^{-1}\left(z_{t-1}^K\right)-\frac{\beta_t}{\sqrt{1-\bar{\alpha_t}}}g_\theta\left(T^{-1}\left(z_{t-1}^K\right),t\right)\right)$
> >
> >
> > $x_{t-1}=\mu_\theta\left(z_{t-1}^K,t\right)+\sigma_t\gamma$
> >
> >
> > where $\gamma\sim\mathcal{N}\left(0,I\right)\mathrm{\ \ if\ }t>1,\ \mathrm{else\ \gamma}=0$. Finally, we can obtain the improved $x_0$.
> >
> >
> > $\textbf{The motivation of the added Experiment:}$
> >
> >
> > The assumption that the noise in $x_t$ is composed of both the noise upon the structural component of $x_0$ and the other noises is supported in section 5.1 IMAGE GENERATION WITH SCDM. If the initial data $x_0$ is contaminated, then $x_t$ actually contains other noises besides the noise upon the structure of $x_0$. We conducted experiments on data contaminated to different degrees and found that this two-step denoising can perform well. In other words, the conventional approach is to get the distribution of the contaminated $x_0$. The distribution learned by SCDM more closely approximates that of the clean $x_0$.
> >
> >
> > $\textbf{Energy conservation formulas for the residuals:}$
> >
> >
> > Our datasets come from Schmidt and Lipson (2009) [1], who collected data from standard experiments used in undergraduate physics education and provided the underlying laws (Hamiltonians and Lagrangians). In the idealized model, each system is conservative. We therefore approximate the corresponding constant energy level by the empirical mean of the Hamiltonian over the training data.
> >
> >
> > $\textbf{Meaning of intrinsic factors of some rules:}$
> >
> >
> > The meaning of intrinsic factors is the structural component of $x_0$ in $x_t$. It is the latent factor SCDM learned. The various expressions of latent factor we used in the paper might have caused you some confusion. We have unified them all as latent factors.
> >
> >
> > Best rewards,
> >
> >
> > ICLR 2026 Conference Submission6110 Authors
> >
> > ---------------------
> > [1] Michael Schmidt and Hod Lipson. Distilling free-form natural laws from experimental data. Science, 324(5923):81–85, April 2009.

---

### Official Review · Reviewer_Qxrv · 2025-10-30

**Soundness:** 3
**Presentation:** 3
**Contribution:** 2
**Rating:** 4
**Confidence:** 3

**Summary:**

This paper proposes a novel reverse diffusion process to force the model to better learn low-dimensional and sparse latent representations. The formula description in the paper is detailed and concise, and experiments have been conducted to verify the effectiveness of this method.

**Strengths:**

- This paper proposes a novel integration of sparse rate reduction with diffusion models which has not been previously explored in the diffusion model literature.
- Tests on 7 physical datasets covering diverse low-dimensional physical systems, with rigorous baselines (DDPM, DDIM, SDD) and three physics-relevant metric.
- The thesis is structured coherently, with clear writing and concise descriptions of the proposed methods. The logical flow of methodology, experiments, and results facilitates easy comprehension of the technical contributions.

**Weaknesses:**

- For quantitative accuracy assessment, the manuscript states that SCDM is compared with 3 baselines (as shown in Table 1), yet Line 346 claims a comparison with 5 baselines—this quantitative discrepancy requires clarification. Furthermore, 3 baselines may be insufficient to fully substantiate the superiority of SCDM, especially given that SCDM does not achieve the best performance across all metrics in Table 1. This limits the persuasiveness of its claimed effectiveness.
- The work proposes a novel reverse process for diffusion models, which in principle could be extended to downstream tasks such as image generation. However, the current study only focuses on sparse compression, with no supplementary experiments to validate SCDM’s performance in other tasks (e.g., image quality evaluation via FID, or model efficiency metrics like inference time/parameter overhead). This narrow scope fails to demonstrate its versatility as a general-purpose diffusion model.
- SCDM’s compression process relies on two core steps: projection and transformation. Ablation studies to isolate and quantify the individual contributions of these two steps are critical to dissecting the model’s working mechanism—without such analyses, the relative importance of each component remains unclear, and the rationality of the proposed pipeline cannot be fully verified.
- The manuscript would benefit from replacing raster graphics with vector graphics (e.g., figure 2). Vector graphics offer higher resolution and scalability, which enhances the clarity and reproducibility of key results—an important aspect of academic manuscript presentation.

**Questions:**

- See Weaknesses.

---

> ### Author Response · Authors · 2025-11-23
>
> Dear Reviewer Qxrv：
>
> Thank you for the detailed and constructive feedbacks on our submission.  Below, we address your concerns and outline our responses to your feedbacks.
>
> Weaknesses:
>
> W1:
>
> We made changes to the manuscript. Specifically, we changed the number of baselines in 5.2 DATA GENERATION WITH SCDM to 4 and added PIDM mentioned in the text to table 2.
>
>
> W2:
>
> We also agree with the limitations of the existing experiments. To further support this point, we have added an experiment on CIFAR10 with noisy training data (Section 5.1 in the revised manuscript).  Specifically, we corrupt the 50k training images with different levels of additive Gaussian noise (Noise = 0%, 30%, 40%, 50%, 60%), while keeping the 10k test images clean. We then train DDPM, SC-VAE, and our SCDM under identical settings and evaluate standard image generation metrics, i.e., FID and Inception Score (IS), as summarized in Table 1. SCDM delivers strong performance across all settings, and its advantage becomes even more pronounced when noise is injected into the training data. We also provide, in the Appendix E, qualitative samples generated under various noise levels and the associated training loss curves.
>
> W3:
>
> We agree that disentangling the contributions of the two core steps in SCDM’s compression pipeline is important. In the revision, we conduct ablations on the 30% noisy CIFAR-10 dataset by isolating (i) SCDM-P : the low-dimensional projection in Eq 13 and (ii) SCDM-I : the ISTA-style sparsification in Eq 15. Both SCDM-P and SCDM-I achieve reasonable performance, but each is consistently inferior to the full SCDM module. The result indicates that neither component alone is sufficient, and that the combined design is necessary to obtain the best generation quality.
>
> W4:
>
> We are very grateful for your detailed feedback on the presentation of the paper's images. We have changed the main image to enhances the clarity and reproducibility of key results.
>
> Please re-evaluate our revised manuscript.
>
> Best regards,
>
> ICLR 2026 Conference Submission6110 Authors

---

### Official Review · Reviewer_cVea · 2025-11-09

**Soundness:** 2
**Presentation:** 2
**Contribution:** 2
**Rating:** 6
**Confidence:** 2

**Summary:**

The authors propose a novel design/regularization procedure incorporated into the noise predictor of a diffusion denoiser. They perform the denoising using a compressed, sparse representation of x_t, referred to as z_t. The compressed, sparse representation is trained to have these properties by following CRATE from Yu et al. 2023. They show that this intervention improves sampling performance on datasets which rightfully lie on a low-dimensional manifold. They also show that these learned representations match the known latent variables of their datasets.

**Strengths:**

Novel idea, good sampling results on their chosen datasets. I cannot speak to the validity of the sparse latent representations aligning with the known latent variables of the physical systems that these diffusion models are modelling.

**Weaknesses:**

I am a bit confused about the construction of their denoising net. Their noise prediction network takes in z_t, the compressed representation of x_t. Shouldn't the noise component of x_t, which the network is trying to predict, be incompressible? So shouldn't z_t be throwing away information that the noise prediction network g needs to do its job? My best guess is that their implementation doesn't actually work this way and they are really predicting x_0 hat instead of epsilon hat, or something like that. Otherwise I have no idea how this achieves reasonable sampling quality. Perhaps this is all fine and I am just terminally confused.

"non-markova"-> non markovian

**Questions:**

Please address my confusion above in the weaknesses section

---

> ### Author Response · Authors · 2025-11-23
>
> Dear Reviewer cVea,
>
> Thank you for the constructive feedback on our submission.
>
> Both in theory and in our implementation, we indeed adopt the standard $\varepsilon$ prediction objective. We restated theory in section 4.1 MODEL FRAMEWORK, and the updated content is marked in blue. Specifically, from the forward process，we can have $\mathbb{E}[\varepsilon \mid x_t]= \dfrac{1}{\sqrt{1-\bar{\alpha}_t}}\Big( x_t - \sqrt{\bar{\alpha}_t}\,\mathbb{E}[x_0 \mid x_t] \Big)$.
>
> $\widehat{\varepsilon_\theta}\left(x_t\right)$ is defined conditionally on $x_t$. The noise prediction network is in fact learning a function that is highly correlated with the posterior mean $\mathbb{E}[x_0\mid x_t]$. Our goal in SCDM is precisely to learn a sparse representation $z_t$ that retains the most essential factors of variation in $x_t$, and to use these factors to predict $\widehat{\varepsilon_\theta}\left(z_t,t\right)$.
>
>
> Best regards,
>
> ICLR 2026 Conference Submission6110 Authors

---

> > ### Comment · Reviewer_cVea · 2025-11-23
> >
> > This response does not resolve my confusion. Do you understand the issue that I'm confused about? Could you paraphrase my question in your own words in a way that makes me say "ah yes, you *do* understand!" ?
> >
> > If you can, that might prevent us from talking in circles around each other.

---

> > > ### Author Response · Authors · 2025-11-24
> > >
> > > Dear Reviewer cVea,
> > >
> > > Thank you for your reply.
> > >
> > > Here is our understanding of your confusion：
> > >
> > > In the standard DDPMs, the denoising network is expected to predict the  Gaussian noise. In our work, after compressing $x_t$ into $z_t$, the information of the original noise may be lost. Therefore, the predicted noise will no longer correspond to the “total” noise present in the original $x_t$. However in our assumption, the noise in x_t is composed of both the noise upon the structural component of $x_0$ and the other noises. We would like to clarify that the two-part noise could be handled separately.
> > >
> > >
> > >
> > > We clarify line 21 in Algorithm 1. Instead of estimating the “total” noise, we explicitly handle the two-part noise by the sparse compression and a noise estimation. The first step applies a sparse compression, which extracts the principal structural directions of $x_t$ and produces $z_{t-1}^K$. In the second step, on the principal directions by the sparse representation, we construct the mean through
> > >
> > >   $\mu_\theta(z_{t-1}^K,t) = \frac{1}{\sqrt{\alpha_t}} \left(
> > > z_{t-1}^K  -\frac{\beta_t}{\sqrt{1-\bar{\alpha}_t}}   g _ \theta (z _ {t-1}^K , t)  \right) $
> > >
> > >
> > > where $g_θ$ predicts the conditional noise$\mathbb{E}[\varepsilon \mid z_{t-1}^K]$. This two-stage filtering allows our SCDM to focus on denoising the structurally meaningful components of $x_0$, so that the improved estimate of $x_0$ can be obtained.
> > >
> > > Best rewards,
> > >
> > > ICLR 2026 Conference Submission6110 Authors

---

> > > > ### Comment · Reviewer_cVea · 2025-11-25
> > > >
> > > > Thanks for your reply. Your latest comment does change my understanding of what's going on. But I still have a lot of questions:
> > > >
> > > > 1. By "clarify" line 21, do you mean "substantially change the meaning of"?? Because it looks like you've replaced $x_t$ (old version of the paper) with $z_{t-1}^K$ (revision). These variables are quite different, right? So which version of this line did you actually use in your implementation? If the old version was a mistake, and by "clarifed" you really mean "corrected", please be clear about that.
> > > >
> > > > 2. The new line 21 seems to say that you're denoising $z_{t-1}^K$ to produce $\mu_\theta$. Line 22 just adds some new random noise to $\mu_theta$ to produce the next $x_{t-1}$. Huh? I thought $z_{t-1}^K$  was supposed to live in a compressed representation space that bore some complex nonlinear relationship to the default data space $x_{t-1}$ lives in. I could see this setup working in practice, but only because your $g_\theta$ network could learn to produce some output which is very much *not* interpretable as an expected noise vector, but instead as something like the residual between $x_{t-1}$ and $z_{t-1}^K$. What would make a lot more sense to me is if there was some decoder network that transformed $mu_\theta$ back from sparse representation space to data space, but your pseudocode doesn't indicate that.
> > > >
> > > > Please provide further clarifications.

---

> > > > > ### Author Response · Authors · 2025-11-26
> > > > >
> > > > > Dear Reviewer cVea,
> > > > >
> > > > > Thank you for your reply.
> > > > >
> > > > > Q1:
> > > > >
> > > > > We apologize for the confusion caused by the previous writing error. It is corrected.
> > > > >
> > > > >
> > > > > Q2:
> > > > >
> > > > > $\sigma_t$ is not a random noise. It follows to the standard sample process [1]:
> > > > >
> > > > > $z\sim\mathcal{N}\left(0,I\right)\mathrm{\ if\ }t>1,\ \mathrm{else\ }z=0$
> > > > >
> > > > >
> > > > > $x_{t-1}=\frac{1}{\sqrt{\alpha_t}}\left(x_t-\frac{\beta_t}{\sqrt{1-\bar{\alpha_t}}}\epsilon_\theta\left(x_t,t\right)\right)+\sigma_tz$
> > > > >
> > > > >
> > > > > Q3:
> > > > >
> > > > > We assume the noise in $x_t$ is composed of both the noise upon the structural component of $x_0$ and the other noises.
> > > > >
> > > > > In our implementation, we use an NN $T$ to encoder $x_t$ to a unified representation space $z_t$ referring to algorithm line 14. In order to extract the principal structural directions of $x_t$, we apply sparse compression in representation space and produce $z_{t-1}^K$ referring to algorithm lines 15 to 18.
> > > > >
> > > > > In order to obtain $x_{t-1}$, we map $z_{t-1}^K$ to same ambient space as $x_t$ through an NN decoder $T^{-1}$. And we refer to the standard sample process (algorithm lines 19 to 22 ) :
> > > > >
> > > > > $\mu_\theta\left(z_{t-1}^K,t\right)=\frac{1}{\sqrt{\alpha_t}}\left(T^{-1}\left(z_{t-1}^K\right)-\frac{\beta_t}{\sqrt{1-\bar{\alpha_t}}}g_\theta\left(T^{-1}\left(z_{t-1}^K\right),t\right)\right)$
> > > > >
> > > > > $x_{t-1}=\mu_\theta\left(z_{t-1}^K,t\right)+\sigma_t\gamma$
> > > > >
> > > > >
> > > > > where $\gamma\sim\mathcal{N}\left(0,I\right)\mathrm{\ \ if\ }t>1,\ \mathrm{else\ \gamma}=0$. Finally, we can obtain the improved $x_0$.
> > > > >
> > > > >
> > > > > The assumption that the noise in $x_t$ is composed of both the noise upon the structural component of $x_0$ and the other noises is supported in section 5.1 IMAGE GENERATION WITH SCDM. If the initial data $x_0$ is contaminated, then $x_t$ actually contains other noises besides the noise upon the structure of $x_0$. We conducted experiments on data contaminated to different degrees and found that this two-step denoising can perform well.
> > > > >
> > > > >
> > > > > Best rewards,
> > > > >
> > > > > ICLR 2026 Conference Submission6110 Authors
> > > > >
> > > > > -----------------------------------------
> > > > > Jonathan Ho, Ajay Jain, and Pieter Abbeel. Denoising diffusion probabilistic models. In Proceedings of the 34th International Conference on Neural Information Processing Systems, NIPS ’20,
> > > > > Red Hook, NY, USA, 2020.

---

> > > > > > ### Comment · Reviewer_cVea · 2025-11-26
> > > > > >
> > > > > > Okay, that's significantly different again!
> > > > > >
> > > > > > So now the sampling procedure is: take x_t, encode it, sparsify it, decode it, denoise it, then finally add more noise back in, now you have x_{t-1}. Does that all sound correct?
> > > > > >
> > > > > > And the training procedure is: take a sample $x_0$, sample some noise $\epsilon$, linearly combine them to get $x_t$, encode/sparsify/decode $x_t$, use *that* to predict epsilon, optimize everything against the MSE of that prediction.
> > > > > >
> > > > > > I am... somewhat surprised that this works. The only difference from my original understanding is that, instead of removing the predicted noise from x_t, you instead remove the predicted noise from this encoded/sparsified/decoded version (let's call it x'_t). This seems like it should only work assuming that a non-obvious property holds:
> > > > > >
> > > > > > $x'_t - E[\epsilon|x'_t] \approx x_t - E[\epsilon|x_t]]$
> > > > > >
> > > > > > $x'_t - x_t \approx E[\epsilon|x'_t] - E[\epsilon|x_t]]$
> > > > > >
> > > > > > I'm still thinking about this property...

---

> > > > > > ### Comment · Reviewer_cVea · 2025-11-26
> > > > > >
> > > > > > Okay, that's significantly different again!
> > > > > >
> > > > > > So now the sampling procedure is: take x_t, encode it, sparsify it, decode it, denoise it, then finally add more noise back in, now you have x_{t-1}. Does that all sound correct?
> > > > > >
> > > > > > And the training procedure is: take a sample $x_0$, sample some noise $\epsilon$, linearly combine them to get $x_t$, encode/sparsify/decode $x_t$, use *that* to predict epsilon, optimize everything against the MSE of that prediction.
> > > > > >
> > > > > > I am... somewhat surprised that this works. The only difference from my original understanding is that, instead of removing the predicted noise from x_t, you instead remove the predicted noise from this encoded/sparsified/decoded version (let's call it x'_t). This seems like it should only work assuming that a non-obvious property holds:
> > > > > >
> > > > > > $x'_t - E[\epsilon|x'_t] \approx x_t - E[\epsilon|x_t]]$
> > > > > >
> > > > > > $x'_t - x_t \approx E[\epsilon|x'_t] - E[\epsilon|x_t]]$
> > > > > >
> > > > > > I'm still thinking about this property...

---

> > > > > > > ### Author Response · Authors · 2025-11-28
> > > > > > >
> > > > > > > Dear Reviewer cVea,
> > > > > > >
> > > > > > > Thank you for your reply.
> > > > > > >
> > > > > > > In our paper, this should be divided into two situations.
> > > > > > >
> > > > > > > First,
> > > > > > >
> > > > > > > $x_t$ in the representation space can be represented by latent factors in a sparse way.  Then  $T^{-1}\left(z_{t-1}^K\right)=x_t$. The model will attempt to generate data that is consistent with the distribution of the input data $x_0$. And $x_t^\prime-E\left[\epsilon\mid x_t^\prime\right]=x_t-E\left[\epsilon\mid x_t\right]$.
> > > > > > >
> > > > > > >
> > > > > > > Second,
> > > > > > >
> > > > > > > $x_t$ in the representation space can not be represented by latent factors in a sparse way. SCDM is a generative model that is implicitly restricted by the learned latent factors. So $x_{t-1}$ in the sampling phase will not be equal to $\sqrt{\bar{\alpha_{t-1}}}x_0+\sqrt{1-\bar{\alpha_{t-1}}}\epsilon$ which is constructed during the forward process. In other words, $x_{t-1}$ will be implicitly restricted by the learned latent factors. And $x_t^\prime-E\left[\epsilon\mid x_t^\prime\right]\neq x_t-E\left[\epsilon\mid x_t\right]$.
> > > > > > >
> > > > > > >
> > > > > > > This can also well explain why when we use the clear images to calculate FID and Inception scores for the images generated on the contaminated CIFAR-10, it is found that SCDM generates images of better quality compared to DDPM.
> > > > > > >
> > > > > > >
> > > > > > > Best rewards,
> > > > > > >
> > > > > > >
> > > > > > > ICLR 2026 Conference Submission6110 Authors

---

### Meta-Review · Area_Chair_jkUm · 2026-01-06

**Summary:**

The paper proposes Sparse-Compression Diffusion Models (SCDM), which integrate a CRATE-style sparse coding mechanism into the reverse steps of a diffusion model to encourage low-dimensional, interpretable representations. Major reviewer concerns include the paper's presentation, the lack of theoretical novelty (viewed as a minor modification of CRATE for the diffusion context), and the limited experimental validation on toy settings and CIFAR-10.

**Reviewer Concerns:**

Concerns regarding the writing and notation remain unresolved. For instance, the claim that the "diffusion process merely overfits in data denoising" is unfounded, as diffusion models are known for their ability to generalize to new images. Furthermore, while the authors provided additional experiments with FID and Inception scores, the evaluation remains limited to small-scale settings like CIFAR-10.

**Reviewer Scores:**

Given that the critiques to paper presentations and empirical validation are not fully addressed, it is highly unlikely that the reviewers g5Ep and otdE would have significantly improved their scores during a discussion.

---

### Decision · Program_Chairs · 2026-01-26

Reject